# STRETCHYSNAKE: FLEXIBLE SSM TRAINING UNLOCKS ACTION RECOGNITION ACROSS SPATIO-TEMPORAL SCALES

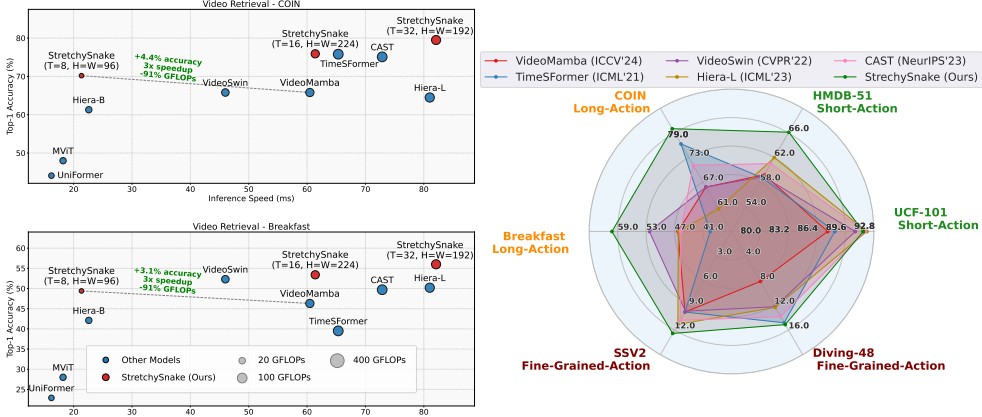

Figure 1: STRETCHYSNAKE: An efficient solution for short- and long-video action understanding in a single model. **Left:** StretchySnake is robust to a wide range of spatial and temporal resolutions, optimizing the accuracy and cost tradeoff depending on the setting. Compared to transformer and SSM baselines, StretchySnake achieves higher accuracy with dramatically lower compute cost. On COIN & Breakfast, StretchySnake achieves up to +4.4% accuracy and >90% GFLOPs reduction over VideoMamba (dashed line). **Right:** Across 6 diverse action recognition benchmarks, spanning short, long, and fine-grained motion, StretchySnake consistently achieves the best retrieval performance, outperforming both transformer-based & SSM counterparts.

## ABSTRACT

State space models (SSMs) have recently emerged as a competitive alternative to transformers in various linguistic and visual tasks. Their linear complexity and hidden-state recurrence make them particularly attractive for modeling long sequences, whereas attention becomes quadratically expensive. However, current training methods for video understanding are tailored towards transformers and fail to fully leverage the unique attributes of SSMs. For example, video models are often trained at a fixed resolution and video length to balance the quadratic scaling of attention cost against performance. Consequently, these models suffer from degraded performance when evaluated on videos with spatial and temporal resolutions unseen during training; a property we call spatio-temporal inflexibility. In the context of action recognition, this severely limits a model's ability to retain performance across both short- and long-form videos. Therefore, we propose a flexible training method that leverages and improves the inherent adaptability of SSMs. Our method samples videos at varying temporal and spatial resolutions during training and dynamically interpolates model weights to accommodate any spatio-temporal scale. This instills our SSM, which we call STRETCHYSNAKE, with spatio-temporal flexibility and enables it to seamlessly handle videos ranging from short, fine-grained clips to long, complex activities. We introduce and compare five different variants of flexible training, and identify the most effective strategy for video SSMs. On 6 action video benchmarks, STRETCHYSNAKE outperforms vanilla VideoMamba by up to 28%, while simultaneously delivering 3x speedups and a 90% reduction in GFLOPs in low-resolution settings. On short-action (UCF-101, HMDB-51) and long-action (COIN, Breakfast) benchmarks, StretchySnake outperforms transformer and SSM baselines alike, with strong adaptability to fine-grained actions (SSV2, Diving-48). Therefore, our method provides a simple drop-in training recipe that makes video SSMs more robust, resolution-agnostic, and efficient across diverse action recognition scenarios.

# 1 INTRODUCTION

Among the broad spectrum of challenges in video action understanding, there are two attributes that are particularly central to what models usually must capture: the length of the action (Choudhury et al., 2024; Yang et al., 2024; Lee et al., 2024b; Kahatapitiya et al., 2024) and its granularity (Zhang et al., 2024b;a; Doughty & Snoek, 2022; Nie et al., 2024; Gupta et al., 2023; 2024). In the context of action length, a robust action recognition model ideally can capture both short, quick actions that only last a few seconds as well as long, complex activities that unfold over minutes. In terms of action granularity, coarse actions (e.g., jumping vs. waving) can often be recognized from global motion patterns, whereas fine-grained actions (e.g., dropping vs. setting down an object) require more attention to subtle spatio-temporal cues; a robust model should also excel at both of these objectives. We refer to this challenge as *unified action recognition*, where a single video backbone should perform well across short-length, long-length, coarse-grained, and fine-grained videos alike.

This challenge is compounded by the fact that action length and granularity require different input characteristics: accurately recognizing long activities often requires many frames over time (Zhou et al., 2025; Ye et al., 2025; Tan et al., 2024), whereas distinguishing fine-grained actions can depend on video length and spatial resolution to capture subtle details (Li et al., 2022; Xu et al., 2024). Furthermore, real-world videos exist at diverse resolutions and durations, ranging from high-resolution sports replays to low-quality surveillance footage. Thus, there is a significant benefit in developing models that adapt seamlessly across diverse frame sizes (spatial resolution) and number of frames (temporal resolution). However, most existing video models are trained in a static fashion (Liu et al., 2022; Arnab et al., 2021; Li et al., 2024; Bertasius et al., 2021), where both the spatial and temporal resolutions are fixed. While this simplifies training, it introduces what we call *spatio-temporal in-flexibility*: a model's accuracy degrades when evaluated at resolutions or clip lengths unseen during training. For example, previous image-only studies have shown that image models suffer massive performance drops when simply tested at spatial resolutions unseen during training (Tian et al., 2023; Beyer et al., 2023). We are the first to show that this traditional method of training still perpetuates inflexibility in both the learned spatial and temporal features (Fig. 3).

Moreover, transformers are the dominant backbone for video understanding tasks, including action recognition, (Siddiqui et al., 2024), object segmentation (Kirillov et al., 2023), and multi-modal learning (Zhu et al.; Lin et al., 2023). Yet, they face two fundamental limitations in the context of unified action recognition, where a single model must efficiently and robustly scale to a variety of spatio-temporal resolutions. First, their quadratic attention complexity makes them computationally prohibitive for extended activities such as long-activity videos. Secondly, their reliance on learning explicit token-to-token relationships usually constrains them to fixed input sizes, preventing generalization across diverse spatio-temporal scales. Consequently, transformer-based video models are ill-suited as a straightforward solution to unified action recognition: we later show that naively applying spatio-temporal flexible training to popular video transformers yields inconsistent, architecture-dependent gains (Sec. A.2).

To this end, state space models (SSMs) (Orvieto et al., 2023; Smith et al., 2022; Gu et al., 2021b) are a promising alternative. Unlike attention, SSMs learn to compress sequences into a hidden state and operate at near-linear complexity, which is advantageous for long-range modeling. Moreover, their recurrence-based formulation is more robust to variable input lengths, which could better support a unified action recognition model that scales from short actions to long activities. However, existing video SSMs (Li et al., 2024) follow traditional training practices with fixed spatio-temporal resolutions, *which underutilizes their context-dynamic, long-range capabilities and forces them to inherit the same rigidity as transformers.*

We address this by proposing a flexible training method tailored to unlock the latent potential of SSMs. Instead of training on a fixed resolution and video length, we dynamically sample different spatio-temporal scales during training and interpolate input size-dependent weights on the fly, without architectural changes. Concretely, we interpolate spatial and temporal positional embeddings and the patch-embedding convolution when sampling different heights, widths, and frames while training (Fig. 2). This instills our model, STRETCHYSNAKE, with spatio-temporal flexibility (or st-flexibility) that (1) generalizes across a wide range of scales, and (2) improves action recognition performance. Because spatio-temporal flexibility can be implemented in multiple ways, we systematically introduce and evaluate five alternative strategies to determine the most effective variant (Sec. 4.2). In summary, we show that spatio-temporal flexibility is both possible and highly effec-

tive in video SSMs, enabling a single backbone to generalize from short clips to long activities, with strong accuracy gains and favorable accuracy-compute trade-offs.

Our main contributions are as follows:

- We introduce the first training paradigm that enhance video SSM's ability to handle arbitrary numbers of frames and resolutions, overcoming the rigidity of existing approaches.
- We systematically explore five strategies for instilling spatio-temporal flexibility, identify the most effective configuration, and train our video SSM STRETCHYSNAKE with the best strategy.
- STRETCHYSNAKE generalizes seamlessly from short clips to long activities, outperforming vanilla VideoMamba by up to +28% in top-1 accuracy across six benchmarks.
- STRETCHYSNAKE's spatio-temporal flexibility enables practical tuning of the accuracy-compute trade-off at inference time without requiring any finetuning (Fig. 1).

## 2 RELATED WORKS

### 2.1 STATE SPACE MODELS

Structured state space models (Gu et al., 2021a;b; 2022a) have shown great promise as efficient and powerful sequencing models in various tasks such as image classification (Zhu et al., 2024; Zheng et al., 2025; Hatamizadeh & Kautz, 2025), video understanding (Li et al., 2024; Zatsarynna et al., 2025), and 3D vision (Jin et al., 2025; Liu et al., 2025; Yoshiyasu et al., 2025) Broadly, their main attraction is their ability to be parameterized as either a convolution or recurrence, enabling GPU compatibility and near-linear scaling complexity with regards to sequence length. Traditionally, SSMs map some time-dependent, continuous input sequence of length $L$ into a latent state representation to predict the evolution of the latent state. Specifically, some input sequence $x(t) \in \mathbb{R}^L$ is mapped to some output sequence $y(t) \in \mathbb{R}^L$ through a learned latent state $h(t) \in \mathbb{R}^N$ of dimensionality $N$. SSMs learn this mapping through a two-stage sequence-to-sequence ordinary differential equation (ODE) consisting of four parameters ($\Delta, \mathbf{A}, \mathbf{B}, \mathbf{C}$):

$$h'(t) = \mathbf{A}h(t) + \mathbf{B}x(t), \tag{1}$$
$$y(t) = \mathbf{C}h(t), \tag{2}$$

where $\mathbf{A} \in \mathbb{R}^{N \times N}$ is the hidden state transition matrix and $\mathbf{B} \in \mathbb{R}^{1 \times N}$ and $\mathbf{C} \in \mathbb{R}^{N \times 1}$ are the input and output projection matrices, respectively. With this being a continuous process, a learnable step size $\Delta$ is introduced to discretize $\mathbf{A}$ and $\mathbf{B}$ with a variety of possibilities (Nguyen et al., 2022; Gu et al., 2022b), but we follow the zero-order hold used in (Gu & Dao, 2023):

$$\bar{\mathbf{A}} = \exp(\Delta\mathbf{A}), \qquad \bar{\mathbf{B}} = (\Delta\mathbf{A})^{-1}(\exp(\Delta\mathbf{A}) - \mathbf{I}) \cdot \Delta\mathbf{B}, \qquad \bar{\mathbf{C}} = \mathbf{C},$$

After discretization, an SSM can be computed either as a linear recurrence (shown on the left) or a global convolution (as shown on the right):

$$h_t = \bar{\mathbf{A}}h_{t-1} + \bar{\mathbf{B}}x_t, \qquad\qquad \bar{\mathbf{K}} = (\bar{\mathbf{C}}\bar{\mathbf{B}}, \bar{\mathbf{C}}\bar{\mathbf{A}}\bar{\mathbf{B}}, \cdots, \bar{\mathbf{C}}\bar{\mathbf{A}}^t\bar{\mathbf{B}}),$$
$$y_t = \bar{\mathbf{C}}h_t, \qquad\qquad y = x * \bar{\mathbf{K}}.$$

Often times, the convolutional parameterization is chosen during training for parallelization, whereas the recurrent parameterization is used during inference for constant-time autoregression.

### 2.2 SPATIO-TEMPORAL CHALLENGES IN ACTION RECOGNITION

The core goal of action recognition is to learn high-quality spatio-temporal features that capture what action is being performed in a video. The length of the action/video has a significant impact on which modeling approaches are most effective, motivating the distinction between 'short-form' (Kuehne et al., 2011; Soomro, 2012) and 'long-form' (Caba Heilbron et al., 2015; Kuehne et al., 2014) action recognition. For example, models tailored for short-form action recognition focus mainly on extracting the best possible representations and reducing information redundancy (Li et al., 2023b; Girdhar et al., 2022; Wang et al., 2023) in short, manually trimmed videos. Conversely, long-form action recognition models have the additional burden of learning long-range dependencies

with efficiency while still retaining the general quality of representations (Bertasius et al., 2021; Yang et al., 2024; Lee et al., 2024a; Bahrami et al., 2023). There is also the additional challenge of modeling fine-grained actions, where two different action classes can be separated by very minute differences. Fine-grained action recognition datasets can span various temporal lengths, such as short, fine-grained actions in (Li et al., 2018) where the direction a diver is facing when entering the water constitutes different classes. There are also fine-grained datasets that focus on long-length actions (Goyal et al., 2017; Pan et al., 2025), requiring subtle yet complex spatio-temporal cues to be recognized and remembered over a long period of time. Thus, achieving strong performance across all these objectives unanimously with a single model is difficult - especially with transformers as previously discussed.

### 2.3 TRAINING DEEP VISION MODELS FLEXIBLY

Some works have explored enabling an image model to generally perform well across multiple resolutions through a variety of means, like changing the model patch sizes/input resolutions (Beyer et al., 2023; Tian et al., 2023; Fan et al., 2024) or aspect ratios (Dehghani et al., 2024) during training. In a similar vein, other works have instilled multi-resolution capabilities in an image model by adopting a multi-stream approach (Xia et al., 2024; Yao et al., 2024; Tian et al., 2023), where training images are resized to different resolutions and simultaneously passed through separate branches to produce multi-scale features. However, these methods requires architectural changes and cannot be used as a drop-in training technique for any model. Tangential works have explored these ideas in the video domain, such as using multiple streams for different temporal resolutions (Zhang et al., 2023), using some method of "choosing" only the important frames or tokens in a video (Zhang et al., 2022; Wang et al., 2025), or some combination of the two (Feichtenhofer et al., 2019). Again, these methods either require architectural changes or are inherently constrained by attention's cost, hence the need for some form of token optimization/reduction.

In contrast, our work takes a simpler and more general approach: we propose st-flexible training that requires no additional branches or architectural changes, but instead adapts the model on-the-fly during training. Our proposed strategy not only enables StretchySnake to handle a wide range of spatial and temporal resolutions - crucially allowing users to later choose the optimal configuration for balancing accuracy and compute cost as highlighted in (Alabdulmohsin et al., 2024) - but also separates itself from previous works in three ways: (1) We adaptively change the model during training without modifying its core structure, (2) we directly change the spatial and temporal input resolutions during training to learn features that generalize across *all* spatio-temporal scales, and (3) we are the first to propose and explore st-flexibility for video SSMs.

## 3 METHODOLOGY

### 3.1 PRELIMINARIES

Consider some video:

$$\mathbf{x} \in \mathbb{R}^{T \times H \times W \times C}, \tag{3}$$

where $(T, H, W, C)$ are the number of frames, height, width, and channels respectively. Typically, video models reduce each frame in a video into a sequence of $N = \frac{H \times W}{P \times P}$ patches: $\mathbf{x}_n \in \mathbb{R}^{(P^2 \times C)}$, where $P$ is a pre-determined patch size such that $(H * W) \equiv 0 \pmod{P}$ and $n \in \{1, \dots N\}$. This process is referred to as *patchification* and is one way to control the amount of compute for video models. After patchification, the spatial embedding $\mathbf{E}_n^s$ is computed for each patch $\mathbf{x}_n$:

$$\mathbf{E}_n^s = \mathrm{conv}(\mathbf{x}_n), \ \mathbf{E}_n^s \in \mathbb{R}^{\frac{H}{P} \times \frac{W}{P} \times D}, \tag{4}$$

where $D$ is the chosen embedding size and $conv(\cdot)$ is either a 2-D or 3-D convolution operation. To account for permutation invariance in transformers and SSMs, a learned spatial positional embedding $\mathbf{E}_{\mathrm{pos}} \in \mathbb{R}^{N \times D}$ is added to each patch embedding (after concatenation) to obtain the final spatial token representation for a single frame $\mathbf{z}^s$:

$$\mathbf{z}^s = (\mathrm{concat}(\{\mathbf{E}_n^s, \ \forall n\}) + \mathbf{E}_{\mathrm{pos})\in \mathbb{R}^{1 \times N \times D}}. \tag{5}$$

This per-frame process must also be applied to the temporal domain in order to be extended to videos. Subsequently, a learnable temporal positional embedding $\mathbf{E}_{\mathrm{temp}} \in \mathbb{R}^{T \times D}$ is added to every

spatial token $\mathbf{z}^s$ for every frame. Thus, the final temporal token representation $\mathbf{z}^t_j \in \mathbb{R}^{1 \times N \times D}$ for a single frame in a video is obtained:

$$\mathbf{z}^t_j = \mathbf{z}^s_j + \mathbf{E}_{\text{temp}[j]}, \quad (6)$$

for all $j \in \{1, \cdots, T\}$. Finally, a classification token $[CLS] \in \mathbb{R}^{1 \times D}$ meant to aggregate the learned information from all patch tokens (Devlin, 2018; Dosovitskiy, 2020) is appended and used for downstream prediction. With the exception of some minor design choices (such as different types of spatio-temporal factorization), virtually every video-based model encodes videos in this manner before learning spatio-temporal representations; this is where we instill st-flexibility (Fig. 2).

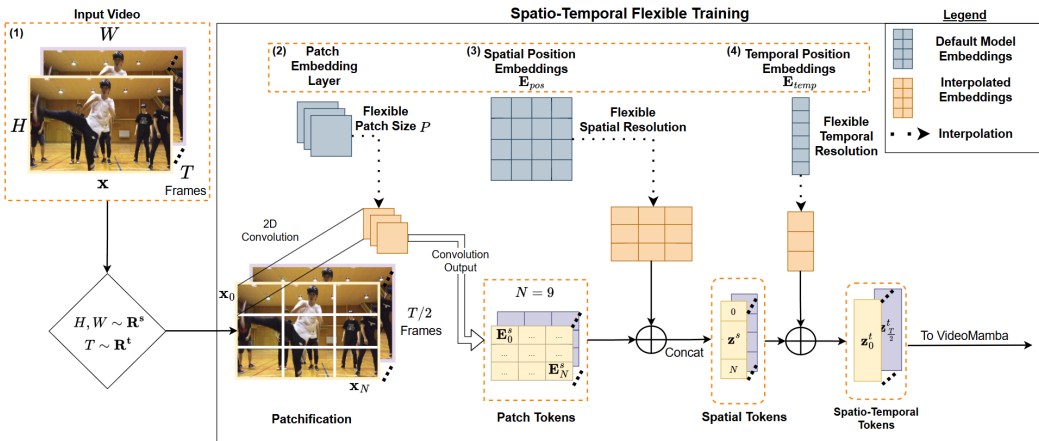

Figure 2: The core idea of spatio-temporal flexibility: we highlight what can be 'flexed' with dashed borders during training. **(1)** The input video can be flexed to a randomly selected spatial $(H, W)$ and temporal $(T)$ resolution. **(2)** The patch embedding layer uses a patch size $P$ which can be flexed to different sizes, changing the outputted number of spatial tokens $N$ for each frame. **(3)** The spatial positional embedding $\mathbf{E}_{\text{pos}}$ must be flexed to accommodate the varying number of spatial tokens $\mathbf{E}^s_N$. **(4)** The temporal positional embeddings $\mathbf{E}_{\text{temp}}$ must be flexed to accommodate the varying number of frames.

## 3.2 MOTIVATION FOR ST-FLEXIBILITY IN SSMs

By instilling *st-flexibility* into video SSMs, we show consistent improvements in their ability to generalize across various spatio-temporal scales and action recognition tasks. While VideoMamba (Li et al., 2024) is the only published video-based SSM model currently available for our work (see Sec. 4.1), st-flexibility is essentially compatible with any future SSM-based video model. This is due to the matrix $\mathbf{A}$ (Eq. 1) in SSMs, which is of particular importance as it is responsible for the latent state-to-state transitions. In other words, it learns to compresses the cumulative history of all previously seen inputs at some timestep into a smaller latent state. It can be difficult to strike a balance between retaining salient information from older context in the model's memory, while still incorporating information from new context - especially so in extremely long contexts. The critical improvement in this facet with SSMs came in (Gu & Dao, 2023) with the selective scan, enabling SSMs to perform content-aware reasoning across long contexts. By simply changing $\mathbf{B}$ and $\mathbf{C}$ to be functions of the input rather than being input-invariant, they can selectively keep or forget information as it propagates through the model.

We hypothesize that this selective retention or forgetting of information (also known as "memory") is a major reason why st-flexibility leads to massive test-time performance improvements specifically in video SSMs. With regards to action recognition, constantly flexing the spatial and temporal resolutions of the video during training encourages the model to learn only the salient action information at a variety of scales. Since $\bar{\mathbf{A}}, \bar{\mathbf{B}}, \bar{\mathbf{C}}$ in VideoMamba depend on the input, we speculate that training VideoMamba with inputs at diverse spatio-temporal scales enhances the memorization of salient information and improves generalization, specifically for SSM-based video models. We discuss this further in Sec. A.7 and provide results on flexibly-trained transformers in Sec. A.2.

### 3.3 INSTILLING ST-FLEXIBILITY

Ideally, a video SSM trained flexibly would generally perform well during test-time across all spatio-temporal scales (low vs. high resolution, short vs. long length, etc.) with minimal drops in performance. Currently, the difficulty in training such a model is two-fold: (1) during training, certain layers and weights in the model must be interpolated accordingly to account for the changes in frame size and video length; and (2) the optimal method of instilling a model with st-flexibility is largely unexplored. Specifically, the convolutional embedding patch size (Eq. 4), number of spatial tokens (Eq. 5), and number of temporal tokens (Eq. 6) are the three key factors that dictate a model's capability to process videos of varying spatial and temporal lengths (Eq. 3). During training, these four equations can be changed (or *flexed*, as we refer to it from here on out) in many different combinations to allow for st-flexibility. In this work, we test 5 different versions of st-flexibility that can be applied to video SSMs during training, which we list below. For all examples, assume the default model expects $T = 16$, $H = W = 224$ as input and $P = 16$ such that $N = \frac{224 \times 224}{16 \times 16} = 196$, $\mathbf{E}_{\text{pos}} \in \mathbb{R}^{196 \times D}$, and $\mathbf{E}_{\text{temp}} \in \mathbb{R}^{16 \times D}$. For st-flexibility, we sample spatial resolutions from the set $\mathbf{R^s} = \{96, 128, 224, 384\}$ and temporal resolutions from the set $\mathbf{R^t} = \{8, 16, 32, 64\}$.

1. **Temporal Flexibility**: Randomly sample $T$ from $\mathbf{R^t}$ during training. Only flex the temporal tokens based on the number of input frames.

   **Example**: If $T \sim \mathcal{U}(\mathbf{R^t})$, assume $T = 32$. Then, $\mathbf{x} \in \mathbb{R}^{32 \times 3 \times 224 \times 224}$, such that $\mathbf{E}_{\text{temp}} \in \mathbb{R}^{16 \times D}$ must be "flexed" to $\mathbf{E}_{\text{temp}} \in \mathbb{R}^{32 \times D}$

2. **Static Patch** - Randomly sample $T$ and $(H, W)$ from $\mathbf{R^t}$ and $\mathbf{R^s}$, respectively, during training. Along with temporal flexibility, image size and number of spatial tokens are flexed, while the patch size is always kept static.

   **Example**: If $(H, W) \sim \mathcal{U}(\mathbf{R^s})$ and $T \sim \mathcal{U}(\mathbf{R^t})$, assume $T = 32$ and $H = W = 128$. Then, $\mathbf{x} \in \mathbb{R}^{32 \times 3 \times 128 \times 128}$ and fix $P = 16$ such that $N = \frac{128 \times 128}{16 \times 16} = 64$ and $\mathbf{E}_{\text{pos}} \in \mathbb{R}^{16 \times D}$ must be "flexed" to $\mathbf{E}_{\text{pos}} \in \mathbb{R}^{64 \times D}$.

3. **Static Tokens**: Randomly sample $T$ and $(H, W)$ during training from $\mathbf{R^t}$ and $\mathbf{R^s}$, respectively. Along with temporal flexibility, image size and patch size are jointly flexed such that the resulting number of spatial tokens for every frame is always the same.

   **Example**: If $(H, W) \sim \mathcal{U}(\mathbf{R^s})$ and $T \sim \mathcal{U}(\mathbf{R^t})$, assume $T = 32$ and $H = W = 128$. If $\mathbf{x} \in \mathbb{R}^{32 \times 3 \times 128 \times 128}$, then $P = 16$ must be "flexed" to $P = 9$ such that $N = \left\lfloor \frac{128}{9} \right\rfloor^2 = 196$ (ensure $N$ is a perfect square) and $\mathbf{E}_{\text{pos}} \in \mathbb{R}^{196 \times D}$ does not need to be "flexed".

4. **FlexiViT**: Introduced in (Beyer et al., 2023) for images, fix $H = W = 240$ and randomly "flex" the patch size and number of spatial tokens from the pre-defined set in the original paper during training. Apply temporal flexing as described in the first example.

   **Example**: If $\mathbf{x} \in \mathbb{R}^{32 \times 3 \times 240 \times 240}$ and $P \sim \mathcal{U}(\{8, 10, 12, 15, 16, 20, 24, 30, 40, 48\})$, assume $P = 12$ such that $N = \frac{240 \times 240}{12 \times 12} = 400$ and $\mathbf{E}_{\text{pos}} \in \mathbb{R}^{196 \times D}$ must be "flexed" to $\mathbf{E}_{\text{pos}} \in \mathbb{R}^{400 \times D}$.

5. **Flex-all**: Randomly sample $T$ and $(H, W)$ from $\mathbf{R^t}$ and $\mathbf{R^s}$, respectively, during training. In addition to image size, convolution kernel size and number of spatial tokens are all flexed during training.

   **Example**: If $(H, W) \sim \mathcal{U}(\mathbf{R^s})$ and $T \sim \mathcal{U}(\mathbf{R^t})$, assume $T = 32$ and $H = W = 128$. Then, $\mathbf{x} \in \mathbb{R}^{32 \times 3 \times 128 \times 128}$, and choose $P$ such that $(0 \equiv P \bmod 128)$ and $12 \leq P \leq 48$. Assume $P = 32$ such that $N = \frac{128 \times 128}{32 \times 32} = 16$ and $\mathbf{E}_{\text{pos}} \in \mathbb{R}^{196 \times D}$ must be "flexed" to $\mathbf{E}_{\text{pos}} \in \mathbb{R}^{16 \times D}$.

To flex the spatial resolution $(H, W)$ of a video we use the Resize function in PyTorch, and to flex the temporal resolution of a video $(T)$, we simply change the number of frames we uniformly sample in a training clip (Eq. 3). To flex the patch size of a model, we resize the weights $w$ of the patch embedding layer (*conv* in Eq. 4) and the spatial positional embedding $\mathbf{E}_{\text{pos}}$ (Eq. 5) to the correct size using a 2-D bi-cubic interpolation. Lastly, we use a simple 1-D linear interpolation to flex the temporal positional embedding $\mathbf{E}_{\text{temp}}$ to the correct size. Since all interpolation operations applied to $w$, $\mathbf{E}_{\text{pos}}$, and $\mathbf{E}_{\text{temp}}$ are differentiable, their weights are updated through backpropagation during st-flexible training.

## 4 EXPERIMENTS AND RESULTS

To validate that st-flexible training leads to better generalized representations, we structure this section into three main objectives: **(1)** identifying the optimal type of st-flexibility (Sec. 4.2), **(2)** demonstrating the substantial performance of StretchySnake over vanilla VideoMamba on unseen data (Sec. 4.3), and **(3)** benchmarking StretchySnake against SOTA action recognition baselines (Sec. 4.4). We conduct experiments across short-video, long-video, and fine-grained action recognition datasets, and evaluate performance using three different protocols: video retrieval, fine-tuning, and linear probing. First, in Table 1 we perform video retrieval at various spatio-temporal scales on four coarse-grained action recognition datasets: two short-video datasets (UCF101 (Soomro, 2012), HMDB-51 (Kuehne et al., 2011)) and two long-video datasets (COIN (Tang et al., 2019), Breakfast (Kuehne et al., 2014)). Next, we extend our analysis to include fine-tuning and linear probing in Fig. 4 and incorporate **two** additional fine-grained datasets: SmthSmthV2 (Goyal et al., 2017) and Diving-48 (Li et al., 2018)). Finally, we compare StretchySnake with previous SOTA video models pre-trained on Kinetics-400 and show that StretchySnake generalizes better on average across all datasets in video retrieval than all other models (Table 2). *Results on additional datasets (Sec. A.4), training transformers flexibly (Sec. A.2), qualitative results (Sec. A.7), and additional ablations can all be found in the appendix.*

### 4.1 IMPLEMENTATION DETAILS

To obtain StretchySnake, we pre-train VideoMamba on Kinetics-400 (Kay et al., 2017) identically to the vanilla configuration, with the sole modification of incorporating the best method of st-flexibility (Sec. 4.2). Specifically, we train with simple cross-entropy loss for $50$ epochs using the AdamW optimizer with $5$ linear warm-up epochs. We use the default learning rate and weight decay values of $1e^{-3}$ and 0.05, respectively. We initialize StretchySnake with the provided self-supervised pre-trained weights on Kinetics-400 (similarly done in (Tian et al., 2023)), and implement st-flexibility when performing further supervised training on Kinetics-400. For temporal flexibility, we arbitrarily chose $\mathbf{R^t} = \{8, 16, 32, 64\}$. For all types of st-flexibility where applicable, we arbitrarily chose $\mathbf{R^s} = \{96, 128, 224, 384\}$. For FlexiViT, we follow their method by fixing $H = W = 240$ and randomly sampling from a set of patch sizes $\{8, 10, 12, 15, 16, 20, 24, 30, 40, 48\}$ during training. All experiments are performed on one NVIDIA A100 80GB GPU.

### 4.2 FINDING THE OPTIMAL ST-FLEXIBILITY

To find the optimal type of st-flexibility for action recognition with SSMs, we perform video retrieval across 4 different action recognition datasets, across different spatial and temporal resolutions. Figure 3 visualizes the results on one short-action dataset (HMDB-51) and one long-action dataset (Breakfast) in the interest of space. We report all results across all four datasets in Table 1, but regardless the visualizations for COIN and UCF-101 can be found in Sec. A.7. We observe that at every temporal resolution and virtually every spatial resolution, static-tokens appears to be the best performing and most robust type of st-flex for video SSMs. For spatial resolutions $< 192$px, static-tokens massively outperforms the next best type of st-flexibility, usually in some range between $1\% - 18\%$. For spatial resolutions $> 192$px, static tokens still either outperforms or is on-par with other st-flexible methods in almost every setting. Importantly to note, not only does every st-flexible method outperform vanilla VideoMamba, as expected, but they also **outperform vanilla VideoMamba at its default configuration of** $T = 16$ **and** $H = W = 224$. *Thus, we conclude that the best type of st-flexibility from our proposed methods is static-tokens, and call the model trained with this best st-flexible method StretchySnake.*

### 4.3 STRETCHYSNAKE BEATS VANILLA VIDEOMAMBA

With static-tokens established as the optimal type of st-flexible method, we directly compare StretchySnake and vanilla VideoMamba's performance on video retrieval in Table 1. StretchySnake beats vanilla VideoMamba at **every** spatial and temporal resolution, both seen and unseen during training, including vanilla VideoMamba's original configuration. Consistent double-digit improvements are observed over vanilla VideoMamba in nearly every setting, across every dataset. The largest improvements on the long-video datasets (COIN and Breakfast) occur at the higher temporal resolutions, due to their specific need for long-context understanding. With the highest average improvement across all datasets being on the 64-frame setting of Breakfast at $24.8\%$, st-flexibility clearly improves VideoMamba's capabilities for long-range understanding. Conversely, the largest improvements with respect to the short-video datasets (UCF101 and HMDB-51) are seen at the

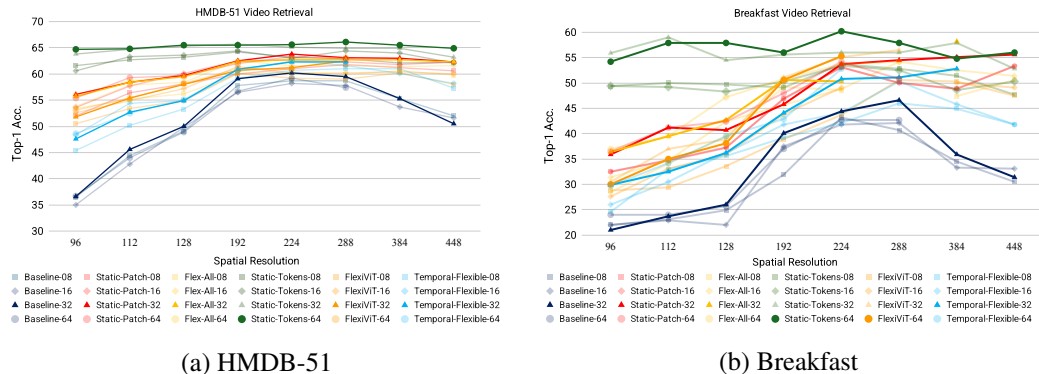

(a) HMDB-51        (b) Breakfast

Figure 3: **Best viewed with zoom.** Video retrieval on HMDB-51 and Breakfast at various spatio-temporal scales (UCF-101 and COIN graphs in Sec. A.7.1). Across every dataset, at virtually every configuration, static-tokens (the green line) is the best performing method of st-flexibility. The suffix (Baseline−08, Baseline−16, etc.) and marker for each label in the legend denotes temporal resolution. For better visibility, only the best temporal setting for each method is bolded.

lower 8-frame and 16-frame temporal resolution scales. Important to note is the relative stability of StretchySnake across all spatial and temporal resolutions alike, as compared to the drastic drops in performance of vanilla VideoMamba across different spatial resolutions. Thus, StretchySnake (and by extension any SSM trained with st-flexibility) is much better equipped to flexibly adapt to any optimal spatio-temporal resolution for any dataset, as opposed to traditionally trained SSMs.

Table 1: Comparing vanilla VideoMamba (VM) and StretchySnake (SS) in video retrieval. Cells highlighted in gray are seen during training, with "$VM_{fx}/SS_{fx}$" denoting temporal resolution used during evaluation. Best VideoMamba results are in red, with StretchySnake best results in green. StretchySnake outperforms VideoMamba in virtually every setting, even at unseen resolutions and length of videos. VideoMamba encounters out-of-memory (OOM) errors at large spatio-temporal resolutions due to its static patch size, while StretchySnake's adaptability prevents this issue.

(a) Breakfast

| Model | Testing Spatial Resolutions | | | | | | | | Avg. Δ% |
|---|---|---|---|---|---|---|---|---|---|
| | 96 | 112 | 128 | 192 | 224 | 288 | 384 | 448 | |
| $VM_{f8}$ | 22.0 | 23.1 | 24.9 | 31.9 | 43.2 | 40.7 | 34.5 | 30.5 | - |
| $SS_{f8}$ | 49.4 | 50.0 | 49.7 | 49.1 | 53.7 | 52.8 | 51.4 | 47.7 | +19.1 |
| $VM_{f16}$ | 22.0 | 22.9 | 22.0 | 37.5 | 41.8 | 42.1 | 33.3 | 33.1 | - |
| $SS_{f16}$ | 49.4 | 49.2 | 48.3 | 50.3 | 53.4 | 52.5 | 48.6 | 50.3 | +18.4 |
| $VM_{f32}$ | 20.6 | 23.7 | 26.0 | 40.1 | 44.4 | 46.6 | 35.9 | 31.4 | - |
| $SS_{f32}$ | 55.9 | 56.0 | 54.5 | 55.6 | 56.0 | 56.0 | 59.0 | 52.8 | +22.1 |
| $VM_{f64}$ | 23.4 | 24.0 | 25.7 | 37.0 | 42.7 | 42.7 | OOM | OOM | - |
| $SS_{f64}$ | 54.2 | 57.9 | 57.9 | 56.0 | 60.2 | 57.9 | 54.8 | 56.0 | +24.8 |

(b) COIN

| Model | Testing Spatial Resolutions | | | | | | | | Avg. Δ% |
|---|---|---|---|---|---|---|---|---|---|
| | 96 | 112 | 128 | 192 | 224 | 288 | 384 | 448 | |
| $VM_{f8}$ | 43.1 | 49.5 | 52.7 | 58.6 | 62.1 | 61.2 | 58.7 | 56.5 | - |
| $SS_{f8}$ | 70.2 | 70.4 | 71.7 | 71.5 | 72.8 | 73.1 | 71.6 | 71.5 | +16.3 |
| $VM_{f16}$ | 50.5 | 55.0 | 57.6 | 62.1 | 64.8 | 64.7 | 61.2 | 58.6 | - |
| $SS_{f16}$ | 74.6 | 74.9 | 74.6 | 75.7 | 75.9 | 75.7 | 75.5 | 74.6 | +13.6 |
| $VM_{f32}$ | 53.0 | 58.6 | 60.0 | 63.5 | 65.4 | 64.7 | 62.4 | 59.8 | - |
| $SS_{f32}$ | 76.9 | 76.5 | 78.9 | 79.5 | 79.0 | 79.4 | 79.2 | 77.8 | +17.5 |
| $VM_{f64}$ | 53.6 | 58.3 | 61.5 | 65.6 | 65.8 | 65.6 | OOM | OOM | - |
| $SS_{f64}$ | 78.8 | 78.8 | 79.2 | 80.0 | 79.5 | 80.0 | 79.5 | 78.9 | +17.7 |

(c) UCF-101

| Model | Testing Spatial Resolutions | | | | | | | | Avg. Δ% |
|---|---|---|---|---|---|---|---|---|---|
| | 96 | 112 | 128 | 192 | 224 | 288 | 384 | 448 | |
| $VM_{f8}$ | 64.7 | 75.4 | 82.2 | 88.7 | 90.2 | 91.0 | 88.2 | 85.7 | - |
| $SS_{f8}$ | 92.4 | 92.4 | 92.7 | 92.7 | 93.4 | 93.1 | 93.0 | 92.8 | +16.8 |
| $VM_{f16}$ | 66.8 | 77.0 | 82.4 | 89.9 | 91.7 | 91.4 | 89.9 | 87.6 | - |
| $SS_{f16}$ | 92.0 | 93.0 | 93.4 | 94.3 | 93.4 | 94.0 | 94.0 | 93.8 | +8.9 |
| $VM_{f32}$ | 68.1 | 77.1 | 82.7 | 89.6 | 91.8 | 91.7 | 90.0 | 86.8 | - |
| $SS_{f32}$ | 92.7 | 93.0 | 93.3 | 93.4 | 93.9 | 94.0 | 94.0 | 94.0 | +8.8 |
| $VM_{f64}$ | 65.8 | 76.4 | 81.3 | 89.5 | 91.5 | 91.2 | OOM | OOM | - |
| $SS_{f64}$ | 93.0 | 93.2 | 93.1 | 93.6 | 94.3 | 94.5 | 93.8 | 94.3 | +11.0 |

(d) HMDB-51

| Model | Testing Spatial Resolutions | | | | | | | | Avg. Δ% |
|---|---|---|---|---|---|---|---|---|---|
| | 96 | 112 | 128 | 192 | 224 | 288 | 384 | 448 | |
| $VM_{f8}$ | 36.5 | 44.4 | 49.1 | 57.8 | 58.7 | 58.7 | 55.3 | 52.0 | - |
| $SS_{f8}$ | 61.6 | 62.7 | 63.2 | 64.2 | 63.2 | 62.9 | 62.1 | 62.2 | +15.3 |
| $VM_{f16}$ | 35.0 | 42.8 | 49.8 | 56.5 | 58.2 | 57.8 | 53.7 | 51.6 | - |
| $SS_{f16}$ | 60.6 | 63.3 | 63.6 | 64.4 | 63.0 | 64.4 | 64.0 | 62.1 | +12.5 |
| $VM_{f32}$ | 36.6 | 45.6 | 50.0 | 59.1 | 60.2 | 59.5 | 55.3 | 50.5 | - |
| $SS_{f32}$ | 63.8 | 64.7 | 65.3 | 65.7 | 65.1 | 64.9 | 64.9 | 63.2 | +12.6 |
| $VM_{f64}$ | 36.7 | 44.0 | 48.9 | 56.7 | 59.2 | 59.0 | OOM | OOM | - |
| $SS_{f64}$ | 64.7 | 64.8 | 65.5 | 65.5 | 65.6 | 66.1 | 65.5 | 64.9 | +11.0 |

In Figure 4, we further compare vanilla VideoMamba and StretchySnake by adding fine-tuning and linear probing experiments. We also add results on two fine-grained (SmthSmthV2 (Goyal et al., 2017), Diving-48 (Li et al., 2018)) action recognition datasets. The linear probing results are another testament to StretchySnake's superior learned representations, as freezing the model and only training a linear classifier still leads to significant improvements across every dataset, with a marginal improvement on HMDB-51. Fine-tuning is a less direct comparison of learned representations since both models are entirely unfrozen and trained using the standard, fixed method of training video models. Despite this, after fine-tuning both models with $T = 16, H = W = 224$ for 30 epochs,

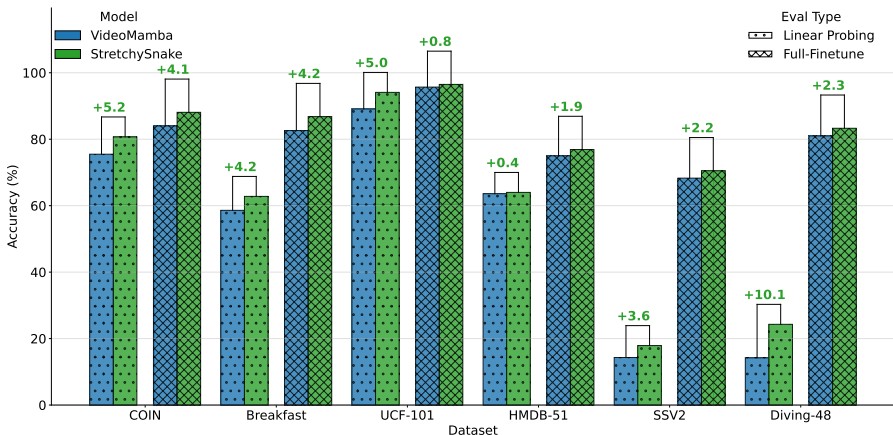

Figure 4: Comparing StretchySnake (green) and VideoMamba (blue) on six different action recognition datasets, both in linear probing and full fine-tuning settings. StretchySnake again outperforms vanilla VideoMamba across every dataset, across both evaluation setting. Thus, st-flexible training enables more robust representations that generalize across various action recognition benchmarks.

StretchySnake's weights serve as a better quality initialization point in this setting as indicated by the uniform improvements across every dataset over vanilla VideoMamba.

## 4.4 STRETCHYSNAKE BEATS SOTA MODELS

Table 2 further compares StretchySnake against current SOTA methods pre-trained on K400 in short-video, long-video, and fine-grained action recognition evaluation. Across six action recognition datasets, StretchySnake performs the best on average, and in some cases outperforms multi-modal models or models trained on extra data. Thus, training VideoMamba with st-flexibility better leverages SSM's dynamic context length modeling capabilities.

Table 2: Video retrieval results with previous SOTA methods trained on K400. StretchySnake massively outperforms vanilla VideoMamba and achieves the best average performance across all datasets. Best **unimodal** results are in green, with second best in red. Gray: models trained on extra modalities (‡) or extra data (†).

| Model | Video Retrieval | | | | | | |
|---|---|---|---|---|---|---|---|
| | UCF101 | HMDB51 | COIN | Breakfast | SSV2 | Diving-48 | Average |
| Uniformer (Li et al., 2023b)(ICLR '22) | 87.4 | 53.4 | 44.1 | 22.9 | 7.8 | 8.3 | 37.3 |
| MViT (Fan et al., 2021)(ICCV '21) | 87.2 | 57.7 | 48.0 | 28.0 | 7.5 | 9.0 | 39.5 |
| Hiera-B (Ryali et al., 2023)(ICML '23) | 94.3 | 62.5 | 61.3 | 42.1 | 11.3 | 9.4 | 47.0 |
| VideoMamba (Li et al., 2024)(ECCV '24) | 91.8 | 60.2 | 65.8 | 46.3 | 9.8 | 8.1 | 47.0 |
| TimeSFormer (Bertasius et al., 2021)(ICML '21) | 91.6 | 58.7 | **76.3** | 39.5 | **11.4** | **14.8** | 48.7 |
| VideoSwin (Liu et al., 2022)(CVPR '22) | 93.9 | 58.9 | 65.8 | **52.3** | 9.7 | 12.2 | 48.8 |
| Hiera-L (Ryali et al., 2023)(ICML '23) | **96.4** | **66.0** | 64.5 | 50.2 | 11.3 | 12.3 | 50.2 |
| CAST (Lee et al., 2024a)(NeurIPS '23) | **95.0** | 65.0 | 75.1 | 49.7 | 11.2 | 13.8 | 51.6 |
| Omnivore (Girdhar et al., 2022)(CVPR '22) † | 95.1 | 62.3 | 71.2 | 53.9 | 10.4 | 9.7 | 50.4 |
| EVL (Lin et al., 2022)(ECCV '22) ‡ | 94.4 | 61.9 | 81.0 | 42.3 | 11.8 | 13.5 | 50.8 |
| UniformerV2 (Li et al., 2023a)(ICCV '23) ‡ | 95.2 | 65.6 | 78.7 | 48.5 | 10.1 | 12.9 | 51.8 |
| FluxViT-S (Wang et al., 2025)(ICCV'25) † | 95.2 | 69.9 | 72.9 | 55.5 | 12.1 | 13.6 | 53.2 |
| AIM (Yang et al., 2023)(ICLR'23) ‡ | 94.5 | 66.0 | 82.8 | 54.2 | 12.5 | 14.0 | 54.0 |
| FluxViT-B (Wang et al., 2025)(ICCV'25) † | 97.0 | 71.1 | 77.3 | 56.4 | 12.5 | 13.7 | 54.6 |
| Ours | 94.5 | **66.1** | **80.0** | **60.2** | **12.4** | **15.1** | **54.7** |

## 5 CONCLUSION

In this paper, we propose a novel method of training video SSMs to instill st-flexibility. During training, we dynamically change the frame size and length of a video to better enable video SSMs to perform well across a vast range of spatial and temporal resolutions. With the variety of combinations with which st-flexibility can be implemented during training, we propose and analyze five different st-flex methods to find the optimal type. Moreover, we show that our model, StretchySnake, achieves SOTA video retrieval performance across six action recognition datasets. With performance gains as high as 28% over vanilla VideoMamba, we effectively demonstrate that StrechySnake contains better quality representations at all spatio-temporal scales; an especially valuable quality given SSM's propensity for learning better long-range dependencies. Additionally, our training method allows for the choice to use any spatial or temporal resolution at inference time without major degradation in performance, accommodating any computational budget.

# 6 REPRODUCIBILITY STATEMENT

Fully anonymized source code is provided in the supplementary material. All st-flexible methods, training configurations and scripts, dataloaders, and evaluation code that pertains to vanilla Video-Mamba and StretchySnake is provided. Code for all SOTA baselines in Table 2 are also present. Most importantly, st-flexible training and every version we propose is thoroughly described in Sec. 3.3 and can easily be reproduced in code following our descriptions.

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

# A APPENDIX

## A.1 OVERVIEW

We organize the appendix into the following sections:

- Sec. A.2 provides results on 3 different transformer models we train flexibly to support our claim that st-flexibility is more compatible with SSMs.
- Sec. A.3 further describes the evaluation protocols used in this paper for clarity.
- Sec. A.4 provides additional video retrieval results on the K700 and LVU datasets.
- Sec. A.5 ablates different patch sizes during evaluation for certain types of st-flexibility which are trained with many different patch sizes.
- Sec. A.6 provides even deeper video retrieval comparisons between StretchySnake and different versions of VideoMamba trained at specific temporal resolutions.
- Lastly, Sec. A.7 provides qualitative results of StretchySnake versus vanilla VideoMamba, namely more [CLS] token and patch activation map visualizations across different datasets and spatial resolutions.

## A.2 TRAINING TRANSFORMERS FLEXIBLY

To further support our claim that st-flexibility is most optimally compatible with SSMs, we train three highly-cited video transformer models (TimeSFormer (Bertasius et al., 2021), UniFormer (Li et al., 2023b), and MViT (Fan et al., 2021)) with the static-tokens st-flex method and provide video retrieval results below (Figures A1 - A12). Notably, both MViT and UniFormer are already designed with some aspect of flexibility - UniFormer uses a rotary positional encoding (RoPE) (Su et al., 2024) which is somewhat robust to variable input sizes, and MViT is a hierarchical multiscale design with pooling attention that can tolerate moderate variation in input resolution. While it may seem that UniFormer and MViT would be perfect candidates for st-flexibility, Figures A1 - A12 shows the high variability in video retrieval results across all models, as opposed to the consistent improvements observed with StretchySnake.

Our conclusions are as follows:

1. TimeSFormer appears to benefit the least from st-flexibility. Across every dataset and spatio-temporal scale, Flex-TimeSFormer performs either on-par, or sometimes worse, than vanilla TimeSFormer. We believe this is due to TimeSFormer's general incompatibility with st-flexibility, as both UniFormer and MViT have already have some aspect of flexibility. Thus, transformers are not as inherently compatible with st-flexible training as SSMs.

2. Meanwhile, UniFormer seems to gain some marginal benefit when trained with st-flexibility, but only at the tail ends of our test spatial resolutions. Flex-Uniformer outperforms vanilla UniFormer at very small and very large resolutions only, while vanilla Uniformer performs better when tested on resolutions close to its training resolution ($224 \times 224$). This is in stark contrast to what we show in the main paper, where StretchySnake outperforms vanilla VideoMamba **even when tested on VideoMamba's deafult spatio-temporal resolution.**

3. Lastly, MViT seems to benefit the best across all transformer models, but the average performance gains of $4 - 10\%$ are nowhere near the massive gains seen between standard VideoMamba and StretchySnake. While we still find this result interesting, it still highlights that st-flexible transformer models are highly architecture dependent, whereas we conjecture that st-flexibility will improve practically any SSM model - this serves as an avenue of potential future work with the availability of additional video-based SSMs.

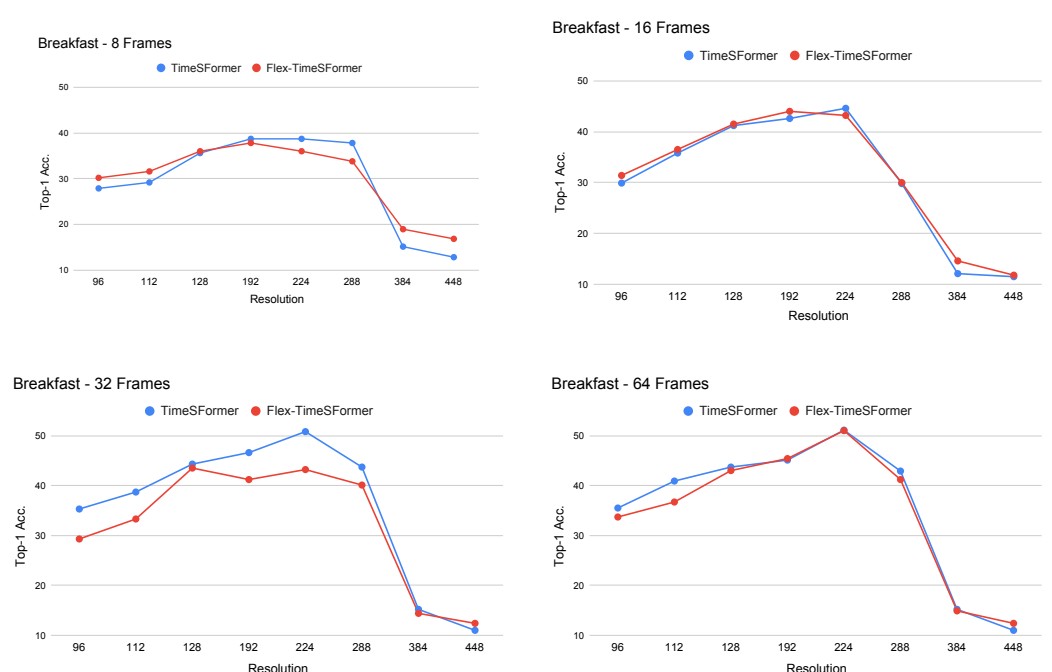

Figure A1: TimeSFormer vs. Flexible TimeSFormer on the Breakfast dataset at various spatio-temporal video retrieval scales.

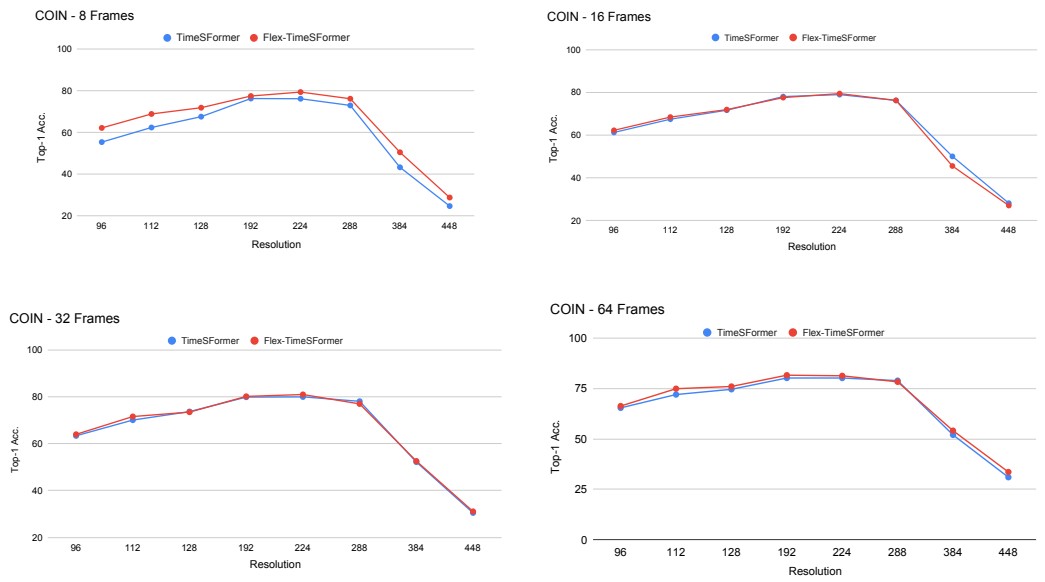

Figure A2: TimeSFormer vs. Flexible TimeSFormer on the COIN dataset at various spatio-temporal video retrieval scales.

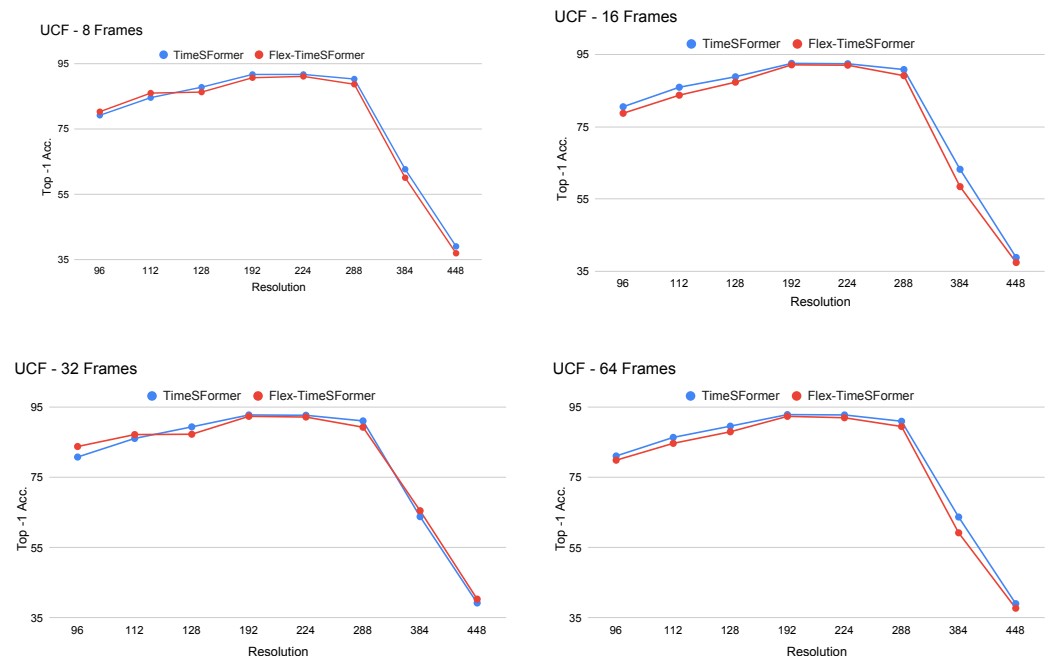

Figure A3: TimeSFormer vs. Flexible TimeSFormer on the UCF-101 dataset at various spatio-temporal video retrieval scales.

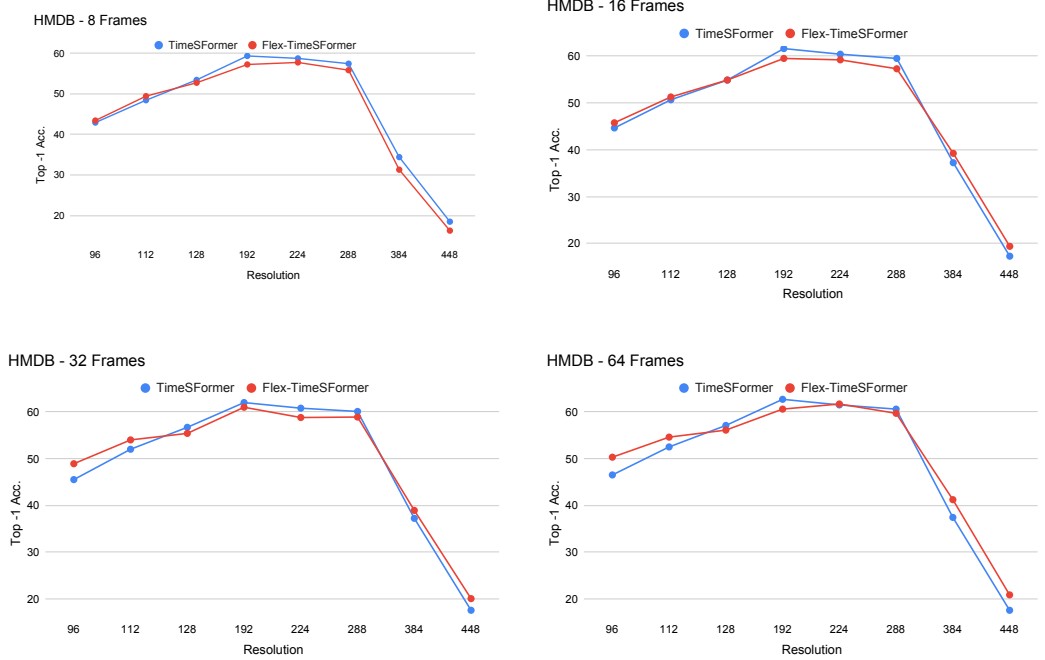

Figure A4: TimeSFormer vs. Flexible TimeSFormer on the HMDB51 dataset at various spatio-temporal video retrieval scales.

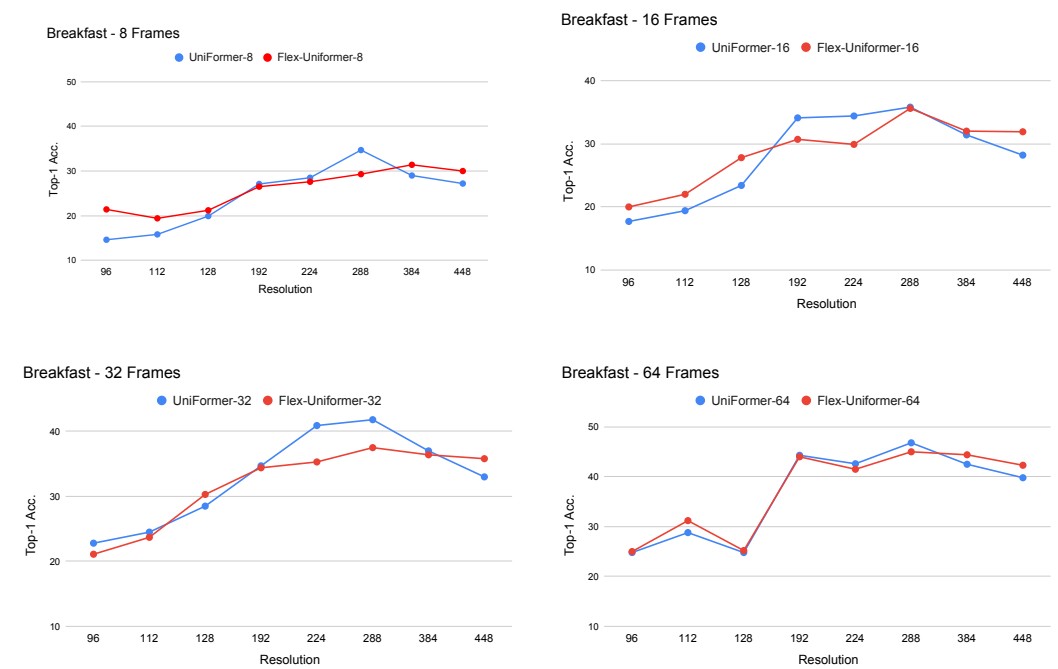

Figure A5: UniFormer vs. Flexible UniFormer on the Breakfast dataset at various spatio-temporal video retrieval scales.

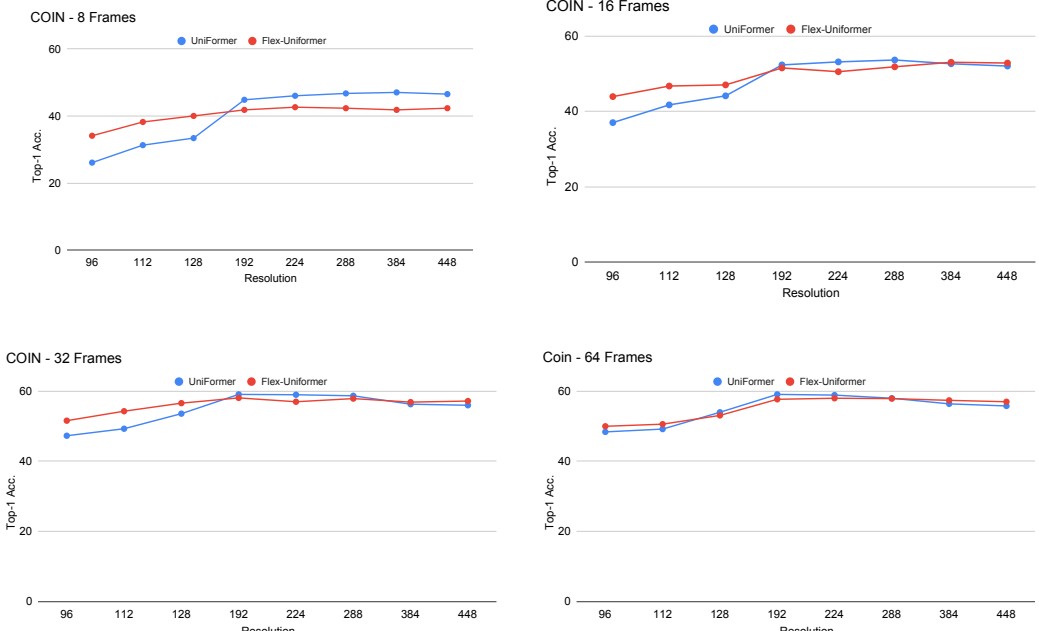

Figure A6: UniFormer vs. Flexible UniFormer on the COIN dataset at various spatio-temporal video retrieval scales.

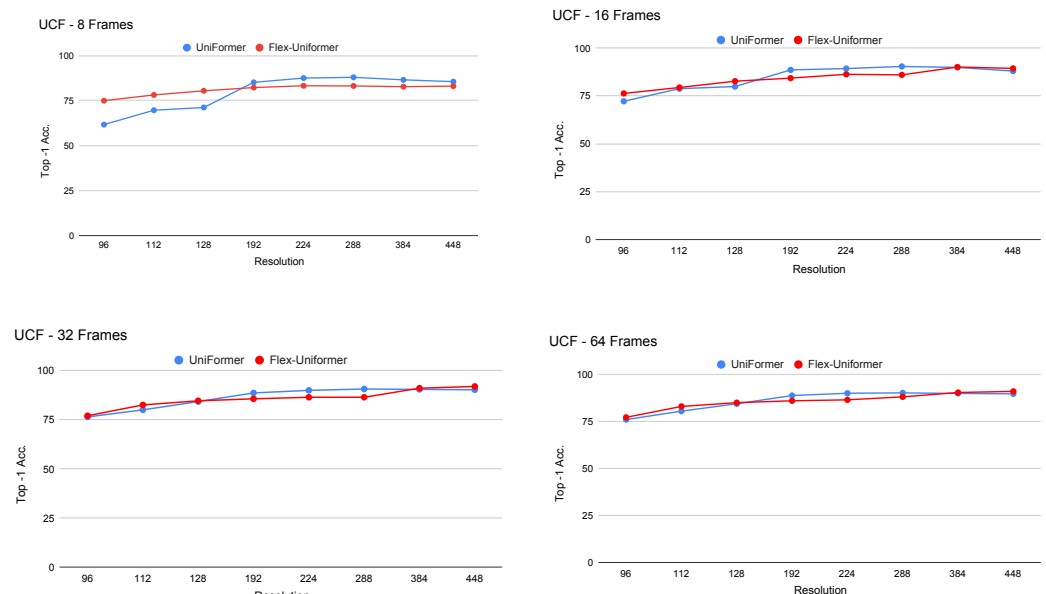

Figure A7: UniFormer vs. Flexible UniFormer on the UCF-101 dataset at various spatio-temporal video retrieval scales.

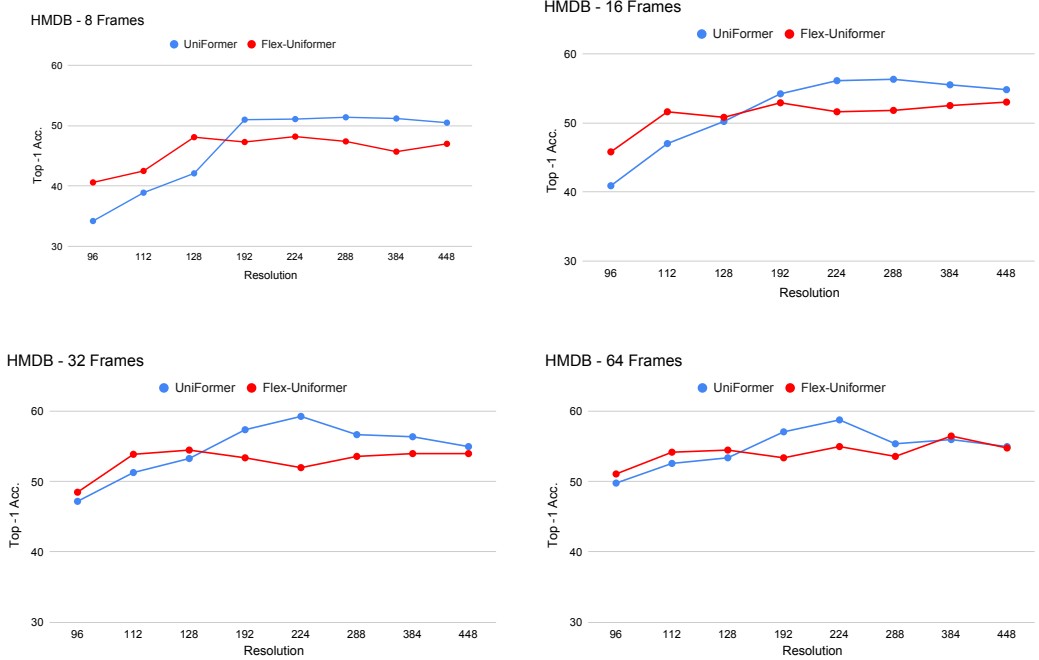

Figure A8: UniFormer vs. Flexible UniFormer on the HMDB51 dataset at various spatio-temporal video retrieval scales.

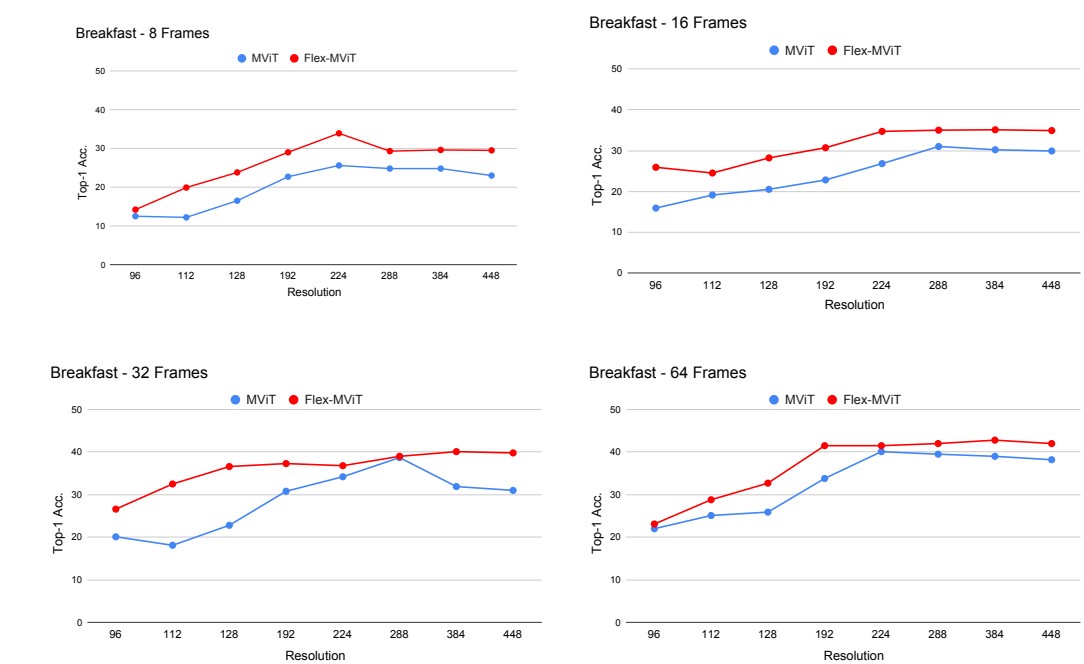

Figure A9: MViT vs. Flexible MViT on the Breakfast dataset at various spatio-temporal video retrieval scales.

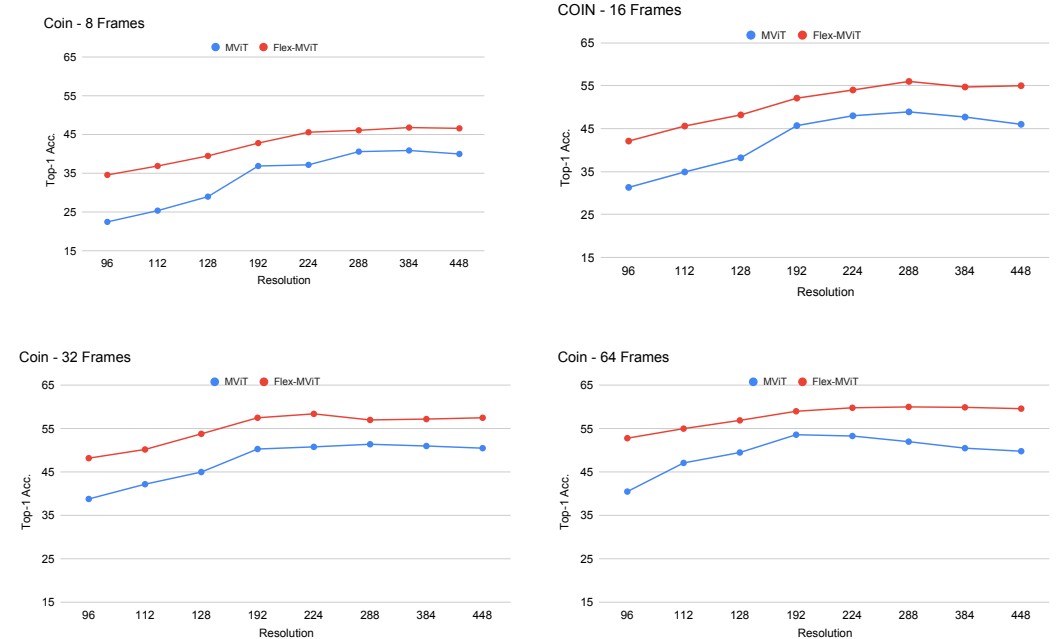

Figure A10: MViT vs. Flexible MViT on the COIN dataset at various spatio-temporal video retrieval scales.

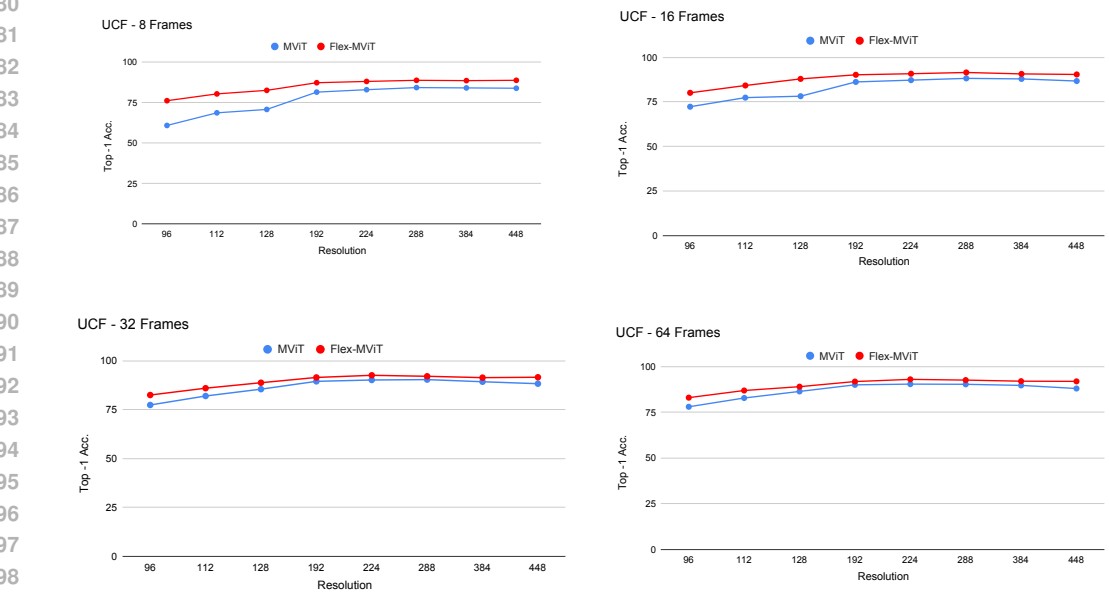

Figure A11: MViT vs. Flexible MViT on the UCF-101 dataset at various spatio-temporal video retrieval scales.

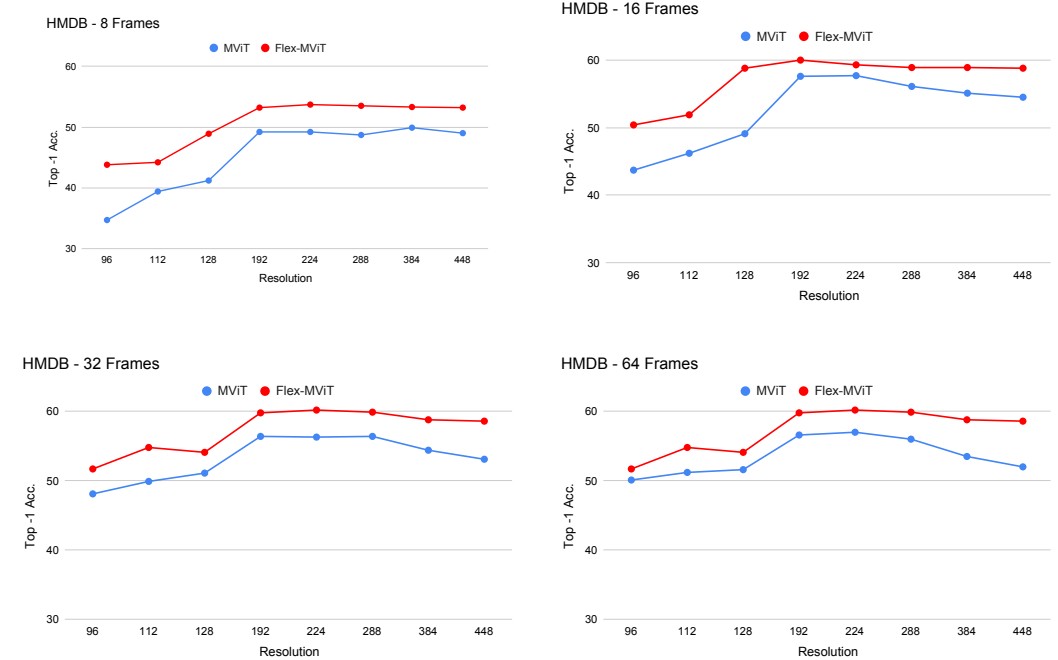

Figure A12: MViT vs. Flexible MViT on the HMDB51 dataset at various spatio-temporal video retrieval scales.

## A.3 Evaluation Details

A majority of our experiments focus on feature evaluation, where the learned representations of our models are frozen and evaluated through a variety of downstream datasets and evaluation protocols. Following previous works such as (Han et al., 2020; Dave et al., 2024; Diba et al., 2021), performing video retrieval and linear probing experiments are effective and appropriate means to evaluate how our proposed st-flexible method leads to better learned representations. For clarity, video retrieval consists of taking all training samples from a dataset and extracting features for each sample (called a gallery) using a pre-trained, frozen model. Then, each test sample (called a query) is passed through the same model and the resulting feature is compared with every feature from the gallery. The nearest neighbor is 'retrieved', and the top-1 accuracy we report throughout the paper is how many test samples and their corresponding top-1 retrieval have the same action label, which is standard protocol.

## A.4 Video Retrieval Results on K700 and LVU

To further validate the effectiveness of our approach, we provide additional video retrieval results on both K700 (Carreira et al., 2019) and LVU (Wu & Krahenbuhl, 2021) in Table A1. Although both StretchySnake and VideoMamba are pre-trained on K400, we report video retrieval results on K700 to provide a comparison on a larger-scale action recognition dataset. Given the substantial overlap in classes between K400 and K700, we restrict evaluation to only classes absent from K400. While K700 is considered a very large-scale action recognition dataset, it is still a short-action dataset with the average video length only being roughly 10 seconds. This evaluation highlights our method's ability to generalize to a significantly larger and more diverse action recognition setting. Additionally, we report results on LVU, which is not an action recognition dataset, but instead emphasizes long-form video understanding across content and metadata tasks. We include results on LVU to further demonstrate StretchySnake's superior ability to adapt to long-form video understanding even outside of action recognition.

Table A1: Video retrieval results comparing StretchySnake and VideoMamba on LVU (content and metadata tasks) and K700. Improvements over VideoMamba are shown in green.

| Model | LVU | | | | | | | K700 (↑) |
|---|---|---|---|---|---|---|---|---|
| | Content (↑) | | | Metadata (↑) | | | | |
| | Rel. | Speak | Scene | Dir. | Genre | Wtr. | Year | |
| VideoMamba | 50.0 | 30.6 | 44.3 | 30.0 | 38.2 | 25.6 | 21.7 | 51.5 |
| StretchySnake | 55.2 (+5.2) | 33.8 (+3.2) | 49.3 (+5.0) | 39.0 (+9.0) | 42.7 (+4.5) | 31.1 (+5.5) | 25.3 (+3.6) | 59.4 (+7.9) |

## A.5 Ablations with Different Patch Sizes for "Flex-All" and "FlexiViT"

An important ablation is examining the effect of patch size during evaluation for the "Flex-All" and "FlexiViT" methods of st-flexibility. Since these methods train with a dynamically changing patch size, we set the patch size $P = 16$ for all experiments in the main paper for fair comparisons to vanilla VideoMamba. However, one could argue that static-tokens is only outperforming these methods due to its adaptive patch size at test time, especially at extremely low or high resolutions. Thus, we provide video retrieval results in Table A2 where we compare static-tokens, Flex-All, and FlexiViT at the same patch sizes. Specifically, we set Flex-All and FlexiViT to use whatever patch size static-tokens would use in the same scenario and show that static-tokens is still the best performing method of st-flexibility.

## A.6 Additional Video Retrieval Comparisons between StretchySnake and VideoMamba

In the main paper, we perform all experiments with a vanilla VideoMamba trained on Kinetics-400 at $T = 16$ and $H = W = 224$. Thus, Table 1 in the main paper evaluates vanilla VideoMamba's performance on unseen spatial AND temporal resolutions. Moreover, since StretchySnake is trained with variable temporal resolutions, it may have an inherit edge against vanilla VideoMamba when we change the temporal resolution in Table 1 of the main paper. Thus, we provide results in Table A3 where we compare StretchySnake with a vanilla VideoMamba trained at the same temporal resolution that is used during evaluation. We simply load the different weights of VideoMamba trained

Table A2: Ablating different patch sizes during evaluation with Flex-All and FlexiViT. Even when using the same patch size as static-tokens, Flex-All and FlexiViT don't reach the same level of performance, further supporting that static-tokens is generally the best method of st-flexible training. Best results are in **bold**.

| Dataset | St-Flexible Method | Testing Spatial Resolutions/Patch Size | | | | | | | |
|---|---|---|---|---|---|---|---|---|---|
| | | 96/7 | 112/8 | 128/9 | 192/14 | 224/16 | 288/21 | 384/27 | 448/32 |
| Breakfast | FlexiViT | 42.6 | 46.0 | 46.3 | 48.3 | 49.1 | 46.3 | 48.3 | 44.9 |
| | Flex-All | 38.7 | 45.4 | 47.0 | 46.9 | 48.8 | 50.2 | 46.0 | 48.6 |
| | Static-Tokens | **49.4** (+6.8) | **49.2** (+3.2) | **48.3** (+1.3) | **50.3** (+2.0) | **53.4** (+4.3) | **52.5** (+2.3) | **48.6** (+0.3) | **50.3** (+1.7) |
| COIN | FlexiViT | 70.7 | 73.6 | 74.4 | 75.2 | 75.7 | 75.5 | 74.7 | 73.6 |
| | Flex-All | 69.4 | 72.3 | 73.8 | 74.9 | 75.5 | 75.3 | 75.4 | 73.9 |
| | Static-Tokens | **74.6** (+3.9) | **74.9** (+1.3) | **74.6** (+0.2) | **75.7** (+0.5) | **75.9** (+0.2) | **75.7** (+0.2) | **75.5** (+0.1) | **74.6** (+0.7) |
| UCF-101 | FlexiViT | 89.4 | 91.2 | 91.3 | 92.3 | 92.5 | 92.4 | 91.7 | 91.3 |
| | Flex-All | 89.0 | 90.5 | 91.4 | 92.3 | 92.4 | 92.4 | 92.0 | 91.8 |
| | Static-Tokens | **92.0** (+2.6) | **93.0** (+1.8) | **93.4** (+2.0) | **94.3** (+2.0) | **93.4** (+0.9) | **94.0** (+1.6) | **94.0** (+2.0) | **93.8** (+2.0) |
| HMDB-51 | FlexiViT | 55.3 | 57.1 | 59.9 | 61.7 | 60.1 | 61.8 | 60.7 | 59.1 |
| | Flex-All | 54.8 | 59.0 | 60.1 | 61.0 | 61.6 | 61.7 | 61.3 | 60.7 |
| | Static-Tokens | **60.6** (+5.3) | **63.3** (+4.3) | **63.6** (+3.5) | **64.4** (+2.7) | **63.0** (+1.4) | **64.4** (+2.6) | **64.0**(+2.7) | **62.1** (+1.6) |

on Kinetics-400 at $T = 8$, $T = 32$, and $T = 64$ originally provided by the authors. For example, we perform video retrieval with $T = 8$ on the Breakfast dataset in rows $1-2$ and compare StretchySnake with a vanilla VideoMamba trained on Kinetics-400 at $T = 8$ and $H = W = 224$. Similarly, we perform video retrieval with $T = 32$ on the Breakfast dataset in rows $3 - 4$ and compare StretchySnake with a vanilla VideoMamba trained on Kinetics-400 at $T = 32$ and $H = W = 224$. Essentially, this is a fairer baseline since we are comparing against vanilla VideoMambas that are performing video retrieval at the same temporal resolution they were trained on. However, StretchySnake still heavily outperforms these models in every setting, further exemplifying StretchySnake's adaptability to any spatio-temporal resolution.

Table A3: Comparing StrechySnake (SS) and vanilla VideoMamba (VM) trained at the same temporal resolution used during evaluation. Cells highlighted in gray are seen during training. Best vanilla VideoMamba results are in red, with best StretchySnake results in green.

(a) Breakfast

| Model | Testing Spatial Resolutions | | | | |
|---|---|---|---|---|---|
| | 112 | 192 | 224 | 288 | 448 |
| VM ($T = 8$) | 17.7 | 38.1 | 35.5 | 42.0 | 26.5 |
| SS ($T = 8$) | **50.0** | **49.1** | **53.7** | **52.8** | **47.7** |
| VM ($T = 32$) | 24.2 | 40.6 | 42.1 | 46.9 | 32.4 |
| SS ($T = 32$) | **56.0** | **55.6** | **56.0** | **56.0** | **52.8** |
| VM ($T = 64$) | 22.0 | 42.9 | 46.8 | 46.8 | 33.2 |
| SS ($T = 64$) | **57.9** | **56.0** | **60.2** | **57.9** | **56.0** |

(b) COIN

| Model | Testing Spatial Resolutions | | | | |
|---|---|---|---|---|---|
| | 112 | 192 | 224 | 288 | 448 |
| VM ($T = 8$) | 50.1 | 60.5 | 65.2 | 63.5 | 58.5 |
| SS ($T = 8$) | **70.4** | **71.5** | **72.8** | **73.1** | **71.5** |
| VM ($T = 32$) | 58.5 | 65.8 | 67.9 | 66.2 | 61.6 |
| SS ($T = 32$) | **76.5** | **79.5** | **79.0** | **79.4** | **77.8** |
| VM ($T = 64$) | 59.9 | 66.1 | 66.4 | 67.6 | 64.3 |
| SS ($T = 64$) | **78.8** | **80.0** | **79.5** | **80.0** | **78.9** |

(c) UCF-101

| Model | Testing Spatial Resolutions | | | | |
|---|---|---|---|---|---|
| | 112 | 192 | 224 | 288 | 448 |
| VM ($T = 8$) | 76.1 | 88.3 | 90.1 | 90.7 | 86.3 |
| SS ($T = 8$) | **92.4** | **92.7** | **93.4** | **93.1** | **92.8** |
| VM ($T = 32$) | 79.5 | 90.0 | 92.5 | 92.4 | 87.7 |
| SS ($T = 32$) | **93.0** | **93.4** | **93.9** | **94.0** | **94.0** |
| VM ($T = 64$) | 80.1 | 90.7 | 92.5 | 92.5 | 89.9 |
| SS ($T = 64$) | **93.2** | **93.6** | **94.3** | **94.5** | **94.3** |

(d) HMDB-51

| Model | Testing Spatial Resolutions | | | | |
|---|---|---|---|---|---|
| | 112 | 192 | 224 | 288 | 448 |
| VM ($T = 8$) | 41.2 | 56.5 | 57.6 | 56.3 | 47.6 |
| SS ($T = 8$) | **62.7** | **64.2** | **63.2** | **62.9** | **62.2** |
| VM ($T = 32$) | 47.1 | 60.8 | 62.7 | 62.5 | 53.4 |
| SS ($T = 32$) | **64.7** | **65.7** | **65.1** | **64.9** | **63.2** |
| VM ($T = 64$) | 47.8 | 61.6 | 62.7 | 62.8 | 59.7 |
| SS ($T = 64$) | **64.8** | **65.5** | **65.6** | **66.1** | **64.9** |

## A.7 QUALITATIVE VISUALIZATIONS

In Fig. 3 of the main paper, we visualize the video retrieval results of all st-flexible methods on the Breakfast and HMDB-51 datasets, and provide the additional visualizations on the UCF-101 and COIN datasets in Sec. A.7.1. We also qualitatively explore StretchySnake at both the classification (Sec. A.7.2) and feature (Sec. A.7.3) levels to visualize its improved representations. The patch

features are the tokens from the last layer that are often discarded since the singular [CLS] token, which is meant to be an aggregation of all patch tokens, is used commonly used for predictions (Bertasius et al., 2021; Dosovitskiy, 2020). However, the final patch features contain more granular information to investigate the spatial activations of a video model at each frame (Oquab et al., 2023). For fairest comparisons to vanilla VideoMamba, in all experiments we fix $T = 16$ and only visualize resolutions that are unseen during training to both StretchySnake and vanilla VideoMamba. We include $H = W = 224$ to show StretchySnake's improvement over vanilla VideoMamba even at the default setting. It is important to note that StretchySnake and vanilla VideoMamba are only trained on the Kinetics-400 dataset, so all visualizations are on completely unseen data.

### A.7.1 COIN AND UCF-101 VISUALIZATIONS

Figure A13 provides video retrieval visualizations of all st-flexible methods on UCF-101 and COIN, similar to the plots on Breakfast and HMDB-51 shown in the main paper. Across both datasets, we observe the same conclusion as the main paper graphs: static-tokens (green curves) achieves the best performance across all spatio-temporal scales. On UCF-101, static-tokens' improvements are most pronounced at lower spatial resolutions, with gains of up to $6\%$ top-1 accuracy over the next best st-flexible method at $96 \times 96$ resolution, and a resounding improvement of over $30\%$ over vanilla VideoMamba. On COIN, the margin is smaller but still clear, with improvements of roughly $+3\%$ at low resolution. As resolution increases, all methods converge toward strong performance, but static-tokens maintains a stable advantage across the all spatio-temporal scales. These results reinforce our conclusion that StretchySnake is able to handle both short-form and long-form retrieval settings while remaining robust across different spatial resolutions.

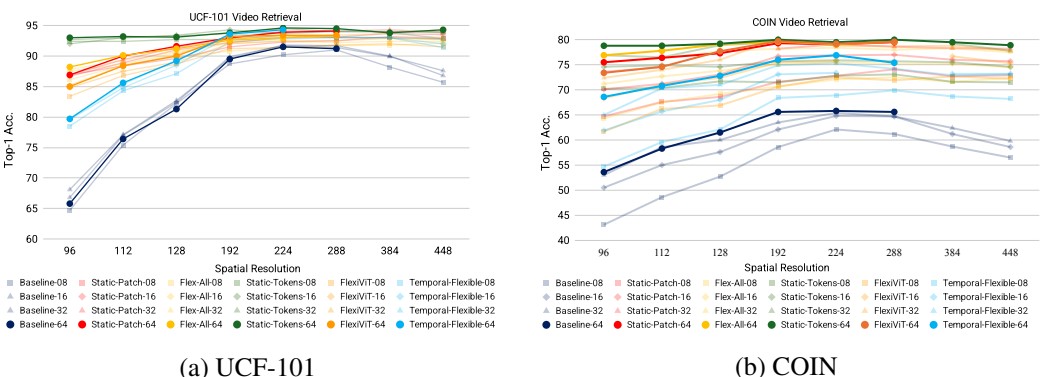

(a) UCF-101  (b) COIN

Figure A13: **Best viewed with zoom.** Video retrieval results on UCF-101 and COIN at various spatio-temporal scales. Across both datasets, at virtually every configuration, static-tokens (the green line) is still the best performing method of st-flexibility. The suffix (Baseline$-08$, Baseline$-16$, etc.) and marker for each label in the legend denotes temporal resolution. For better visibility, only the best temporal setting for each method is bolded.

### A.7.2 [CLS] T-SNE

We provide [CLS] token visualizations at different spatial resolutions during video retrieval on one short action recognition dataset (UCF-101, Figs. A14-A17) and one long action recognition dataset (COIN, Figs. A18-A21). On the UCF-101 dataset (Figs. A14-A17), StretchySnake produces stable, consistent features across all spatial resolution scales, whereas VideoMamba not only struggles to cluster each action at low spatial resolutions, but still does not achieve the same level of clustering as StrechySnake even at high spatial resolutions ($H = W = 448$). On the COIN dataset (Figs. A18-A21), since both models do not achieve the same high levels of accuracy as UCF-101 ($> 90\%$), there is significantly more noise in the visualizations. Despite this, these visualizations serve to similarly show StrechySnake's more stable performance across a variety of unseen spatial resolutions when compared to vanilla VideoMamba. For example, vanilla VideoMamba has considerably more inter-class variation than StretchySnake, specifically at the unseen spatial resolutions ($H = W = \{112, 192, 448\}$).

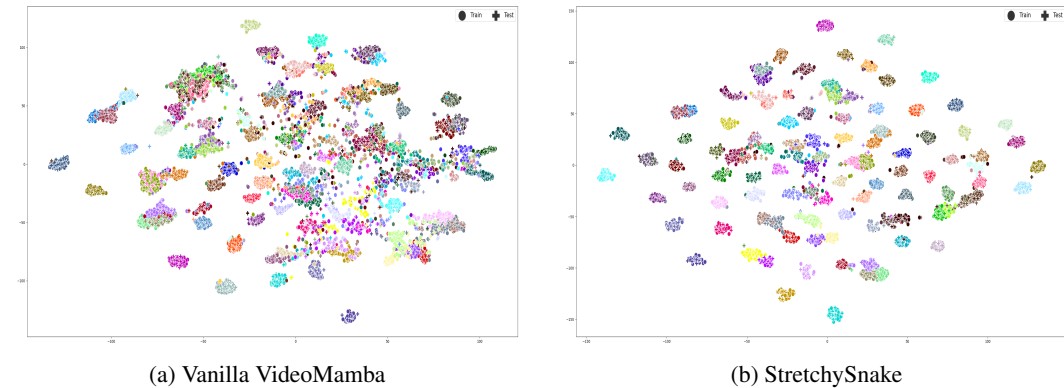

(a) Vanilla VideoMamba                    (b) StretchySnake

Figure A14: [CLS] token visualization on the UCF-101 dataset at $H = W = 112$ pixels. Each color denotes one class (with some redundancy due to the high number of classes). Much tighter clustering of classes indicates StretchySnake's features are more accurate for video retrieval.

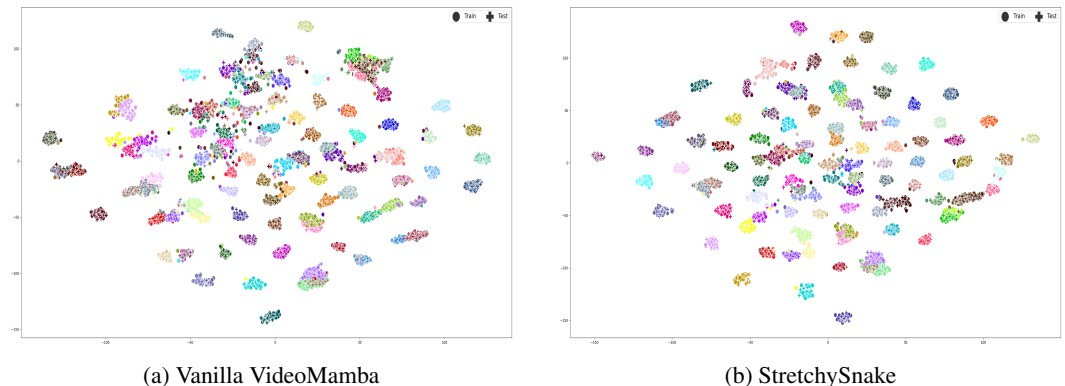

(a) Vanilla VideoMamba                    (b) StretchySnake

Figure A15: [CLS] token visualization on the UCF-101 dataset at $H = W = 192$ pixels. Each color denotes one class (with some redundancy due to the high number of classes). Much tighter clustering of classes indicates StretchySnake's features are more accurate for video retrieval.

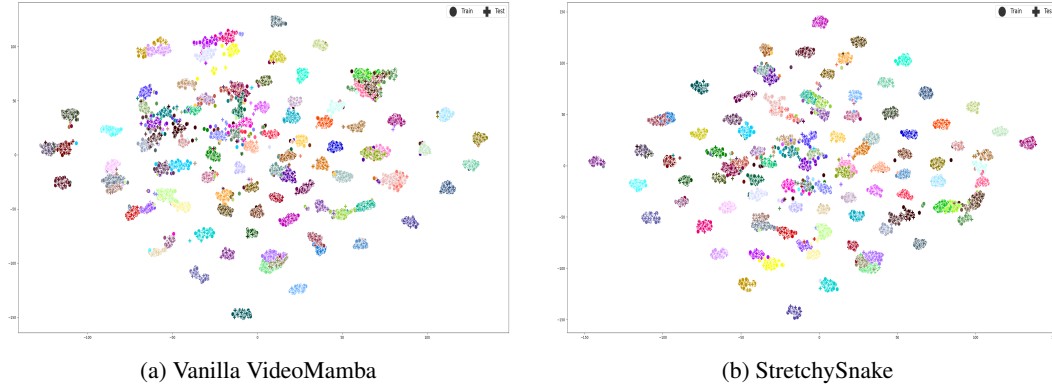

(a) Vanilla VideoMamba                    (b) StretchySnake

Figure A16: [CLS] token visualization on the UCF-101 dataset at $H = W = 224$ pixels. Each color denotes one class (with some redundancy due to the high number of classes). Much tighter clustering of classes indicates StretchySnake's features are more accurate for video retrieval.

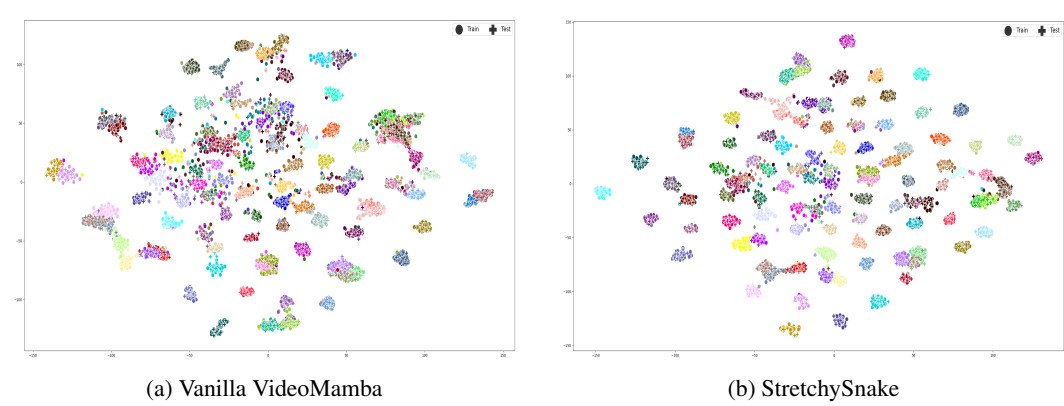

(a) Vanilla VideoMamba  (b) StretchySnake

Figure A17: [CLS] token visualization on the UCF-101 dataset at $H = W = 448$ pixels. Each color denotes one class (with some redundancy due to the high number of classes). Much tighter clustering of classes indicates StretchySnake's features are more accurate for video retrieval.

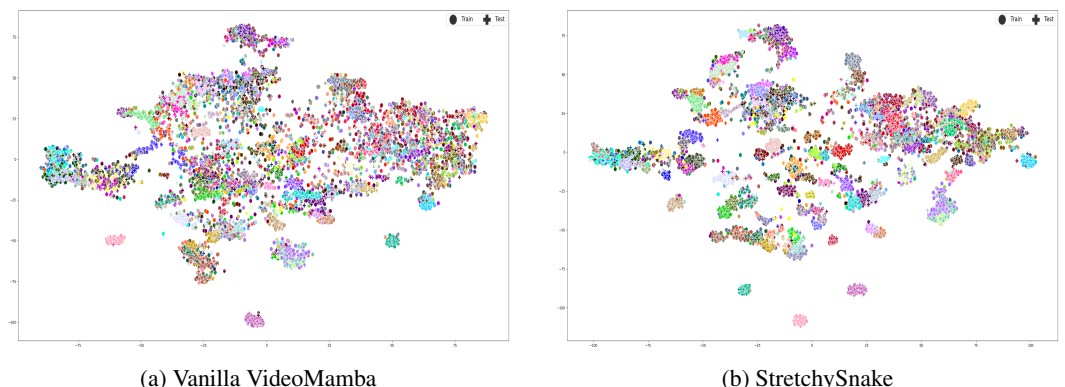

(a) Vanilla VideoMamba  (b) StretchySnake

Figure A18: [CLS] token visualization on the COIN dataset at $H = W = 112$ pixels. Each color denotes one class (with some redundancy due to the high number of classes). Much tighter clustering of classes indicates StretchySnake's features are more accurate for video retrieval.

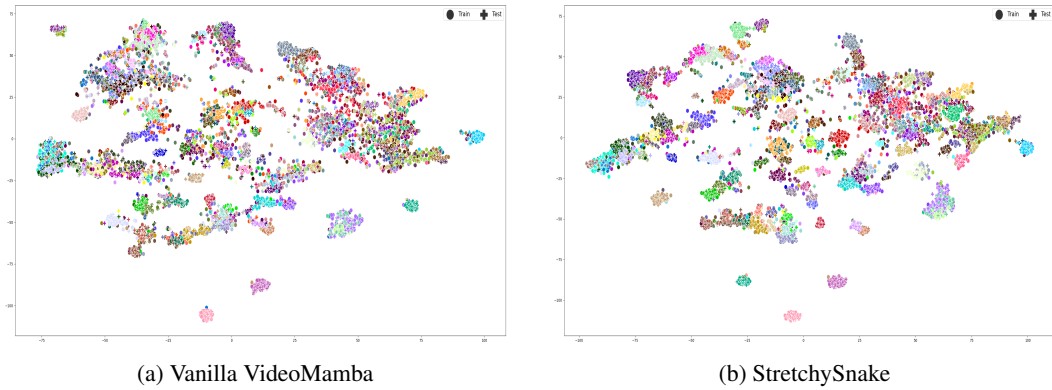

(a) Vanilla VideoMamba  (b) StretchySnake

Figure A19: [CLS] token visualization on the COIN dataset at $H = W = 192$ pixels. Each color denotes one class (with some redundancy due to the high number of classes). Much tighter clustering of classes indicates StretchySnake's features are more accurate for video retrieval.

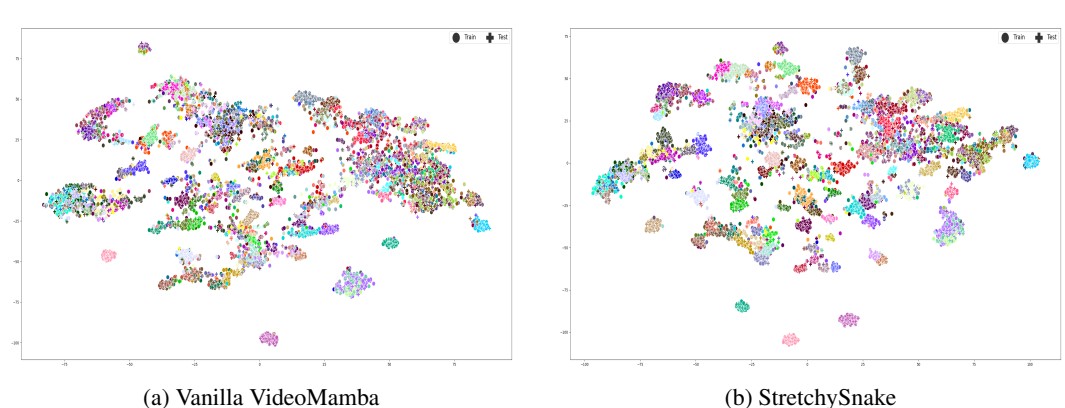

(a) Vanilla VideoMamba  (b) StretchySnake

Figure A20: [CLS] token visualization on the COIN dataset at $H = W = 224$ pixels. Each color denotes one class (with some redundancy due to the high number of classes). Much tighter clustering of classes indicates StretchySnake's features are more accurate for video retrieval.

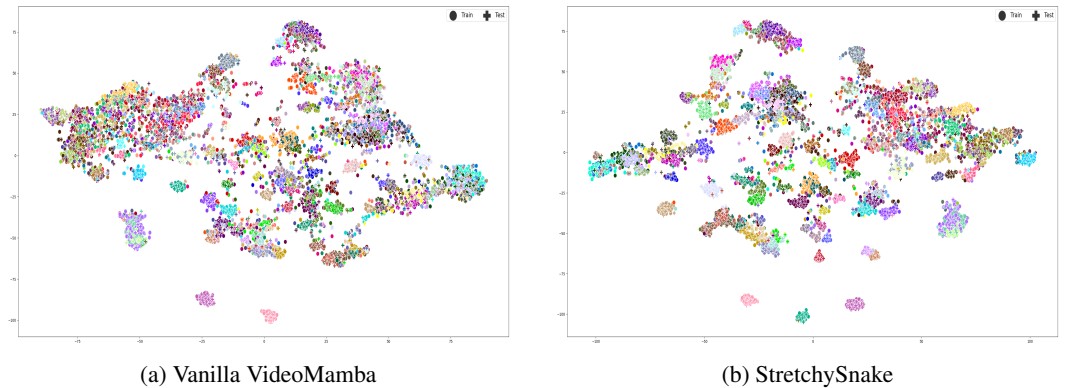

(a) Vanilla VideoMamba  (b) StretchySnake

Figure A21: [CLS] token visualization on the COIN dataset at $H = W = 448$ pixels. Each color denotes one class (with some redundancy due to the high number of classes). Much tighter clustering of classes indicates StretchySnake's features are more accurate for video retrieval.

### A.7.3 PATCH ACTIVATION MAPS

We also provide patch activation maps at different spatial resolutions during video retrieval across the HMDB51 (Figs. A22-A25), COIN (Figs. A26-A29), Breakfast (Figs. A30-A33), and UCF-101 (Figs. A34-A37) datasets. Each graph best viewed with zoom to see finer details. We sample every other frame in the interest of space, and remove black frames which are present in some videos of the COIN dataset.

In the HMDB51 activation maps, StretchySnake exhibits impressive abilities such as tracking and activating on faces, even when they change (left) and on complex and fast moving objects (middle, right). This behavior tracks across all spatial resolutions, whereas vanilla VideoMamba struggles to activate on the correct region at lower resolutions (Figs. A22 and A23) while having difficulty *focusing* on the correct region at higher resolutions (Figs. A24 and A25). Similar behavior is seen on the COIN visualizations, with StretchySnake correctly tracking faces and objects whereas vanilla VideoMamba has either uniformly low activations at low resolutions (Figs. A26 and A27) or random activations at higher resolutions (Figs. A28 and A29). Furthermore, the Breakfast activation maps (Figs. A30-A33) highlight how StretchySnake activates on pertinent items for action recognition (like the coffee mug in the left and middle examples, and the eggs and pan in the right example).

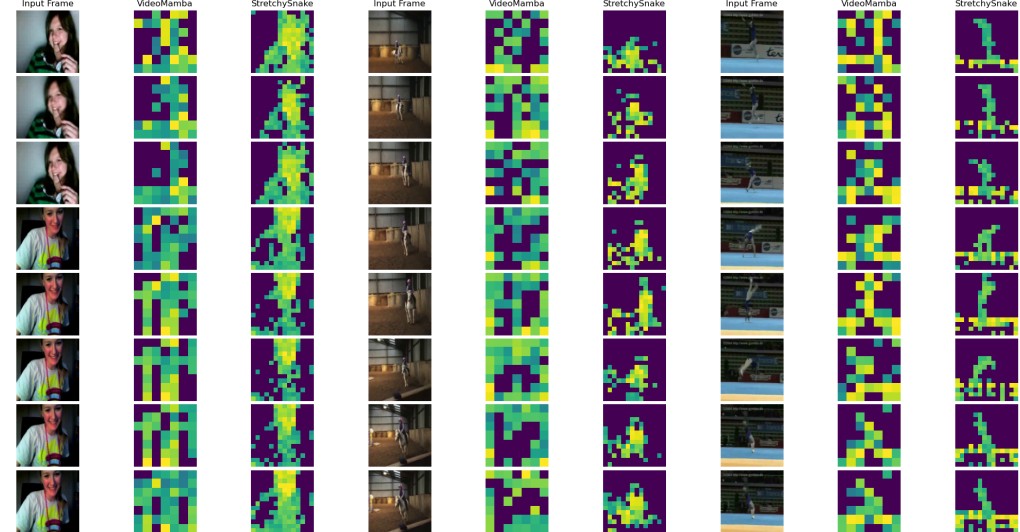

Figure A22: Patch activation map on the HMDB51 dataset at $H = W = 112$ pixels.

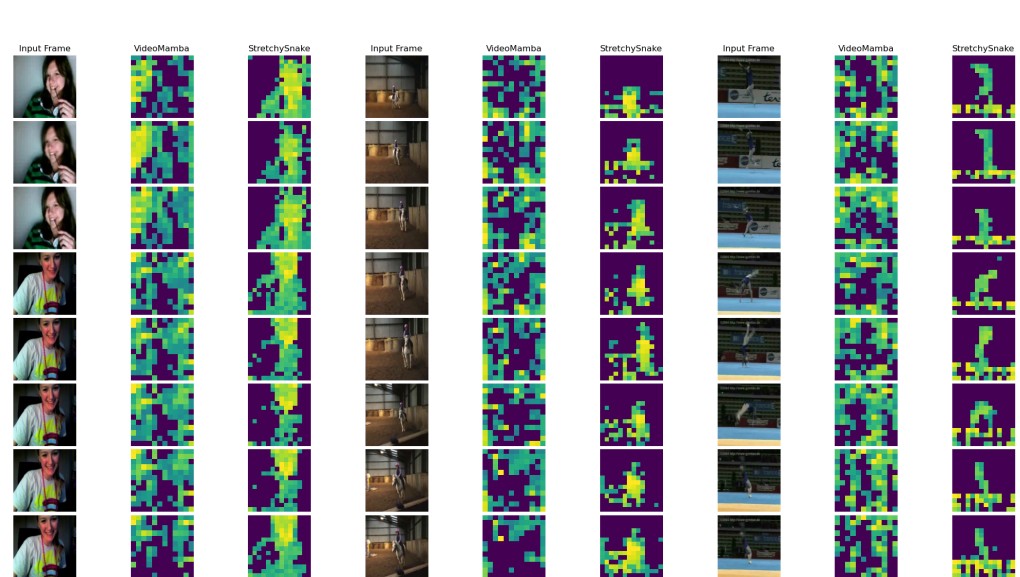

Figure A23: Patch activation map on the HMDB51 dataset at $H = W = 192$ pixels.

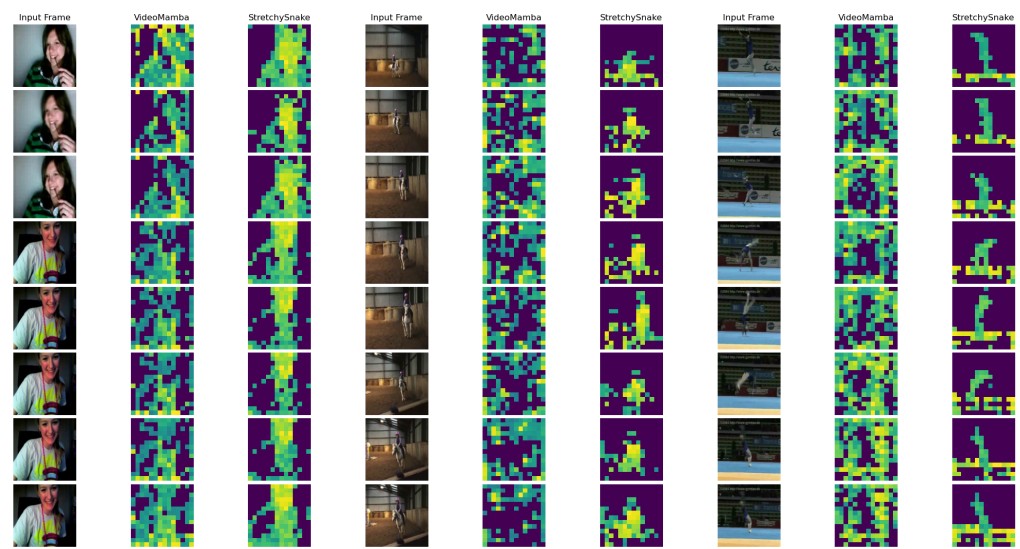

Figure A24: Patch activation map on the HMDB51 dataset at $H = W = 224$ pixels.

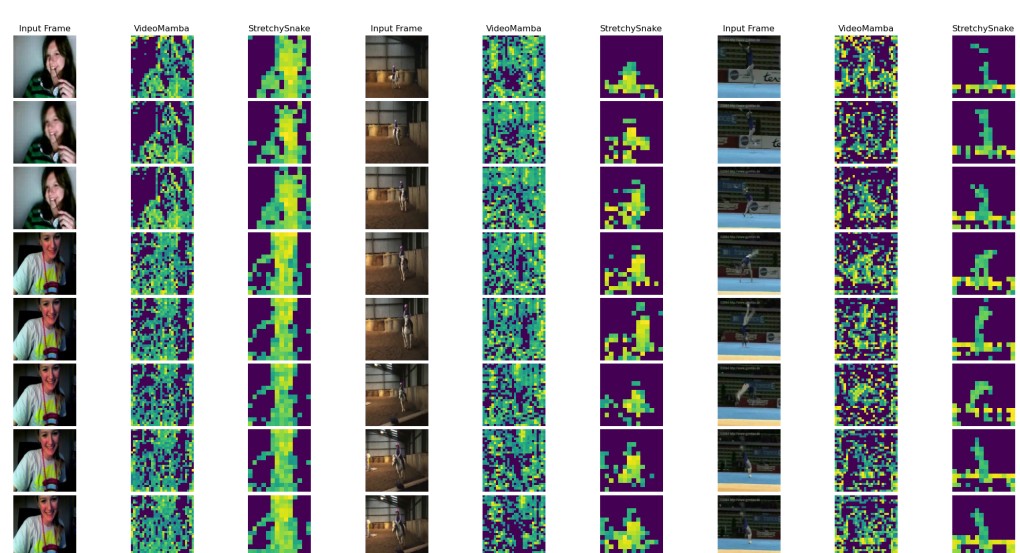

Figure A25: Patch activation map on the HMDB51 dataset at $H = W = 448$ pixels.

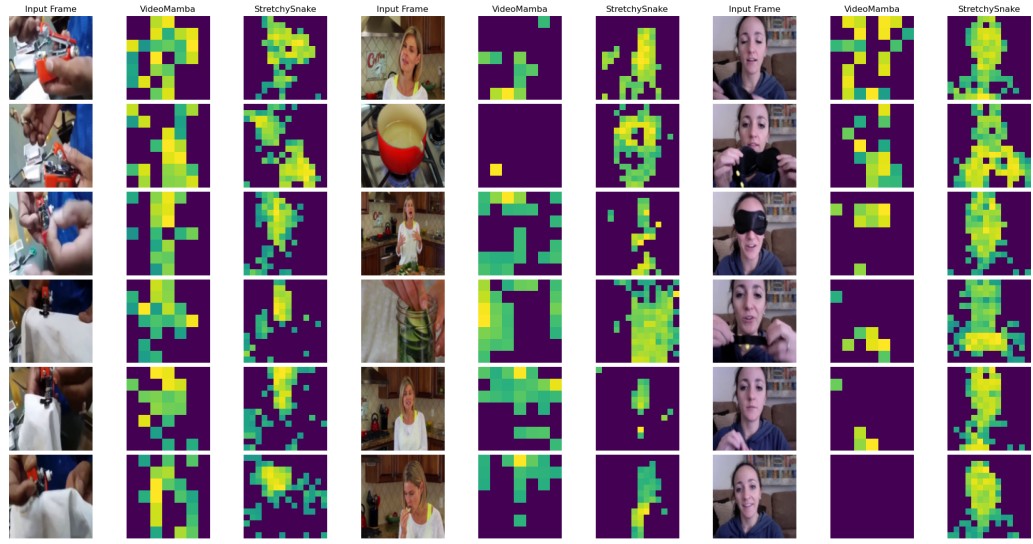

Figure A26: Patch activation map on the COIN dataset at $H = W = 112$ pixels.

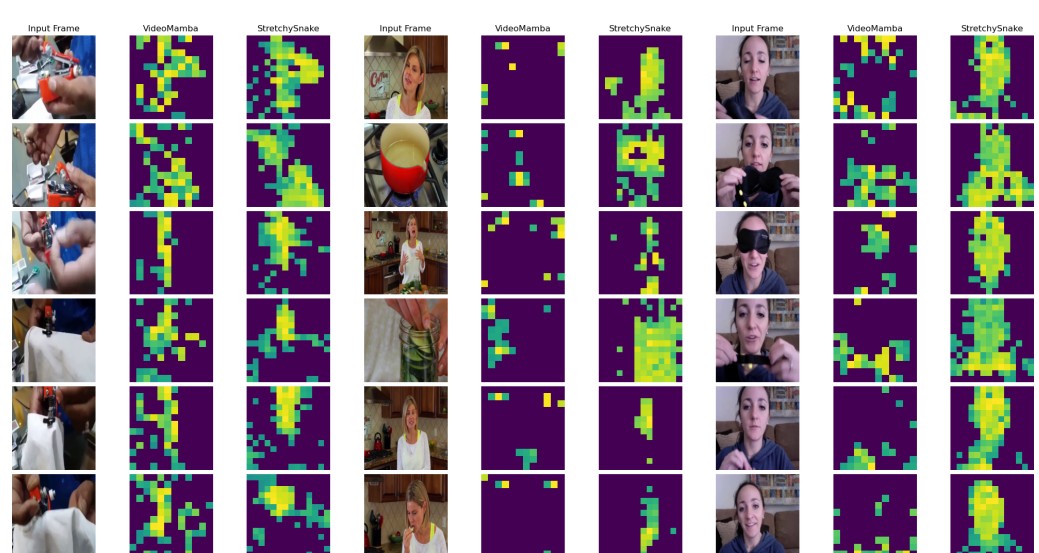

Figure A27: Patch activation map on the COIN dataset at $H = W = 192$ pixels.

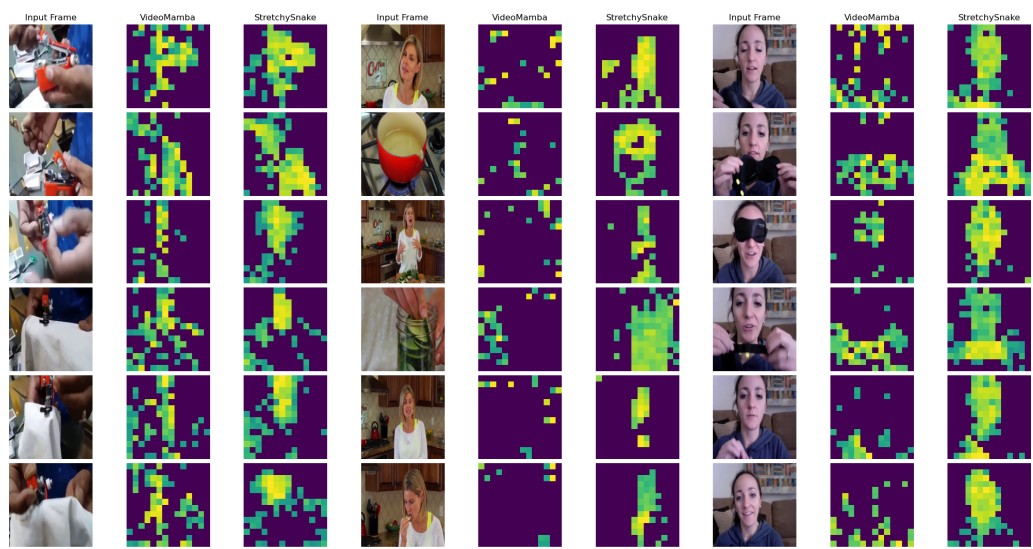

Figure A28: Patch activation map on the COIN dataset at $H = W = 224$ pixels.

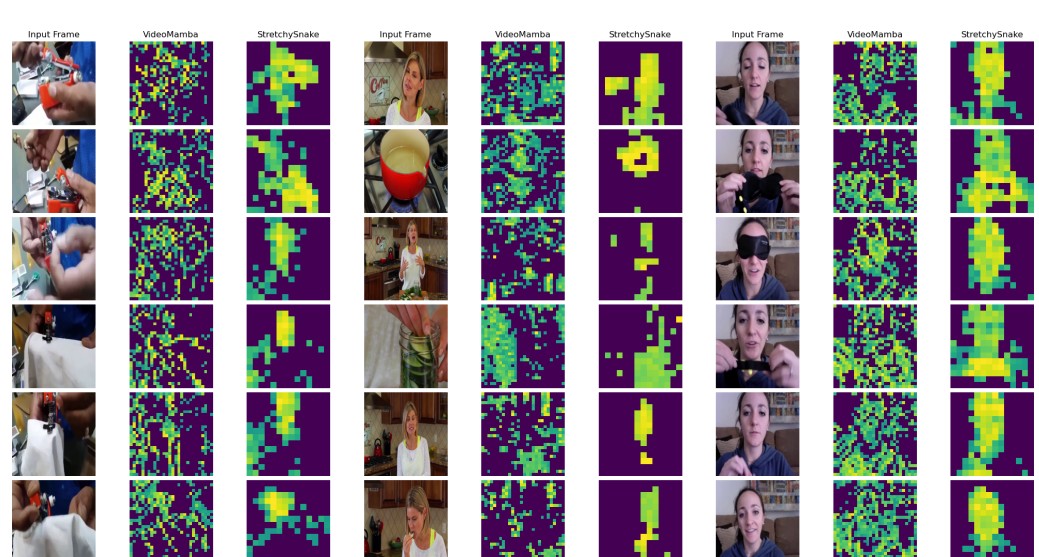

Figure A29: Patch activation map on the COIN dataset at $H = W = 448$ pixels.

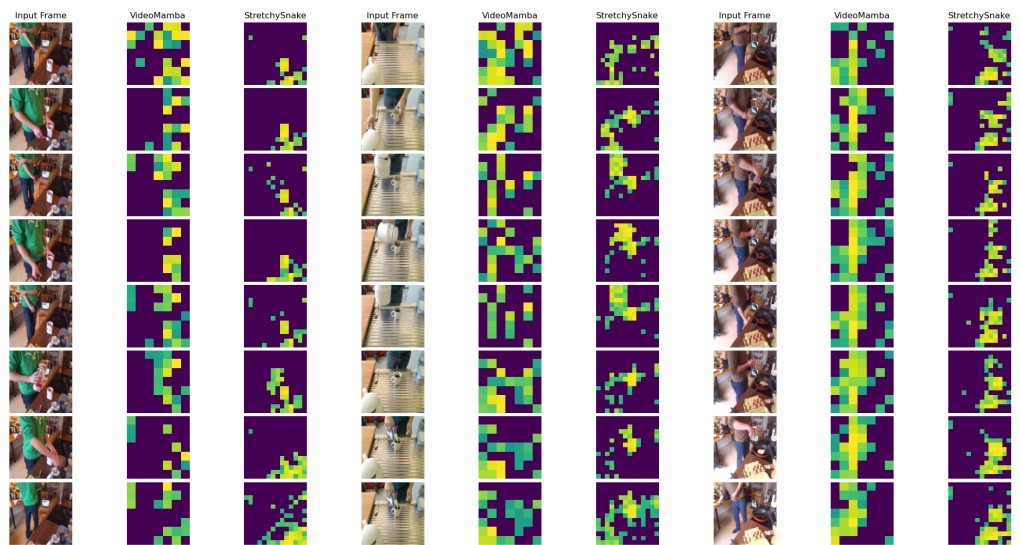

Figure A30: Patch activation map on the Breakfast dataset at $H = W = 112$ pixels.

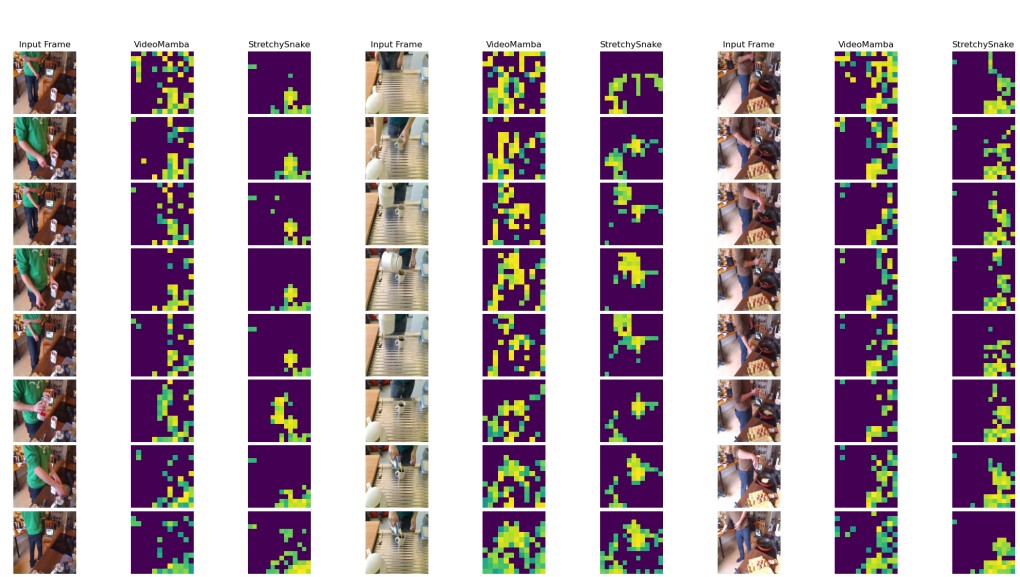

Figure A31: Patch activation map on the Breakfast dataset at $H = W = 192$ pixels.

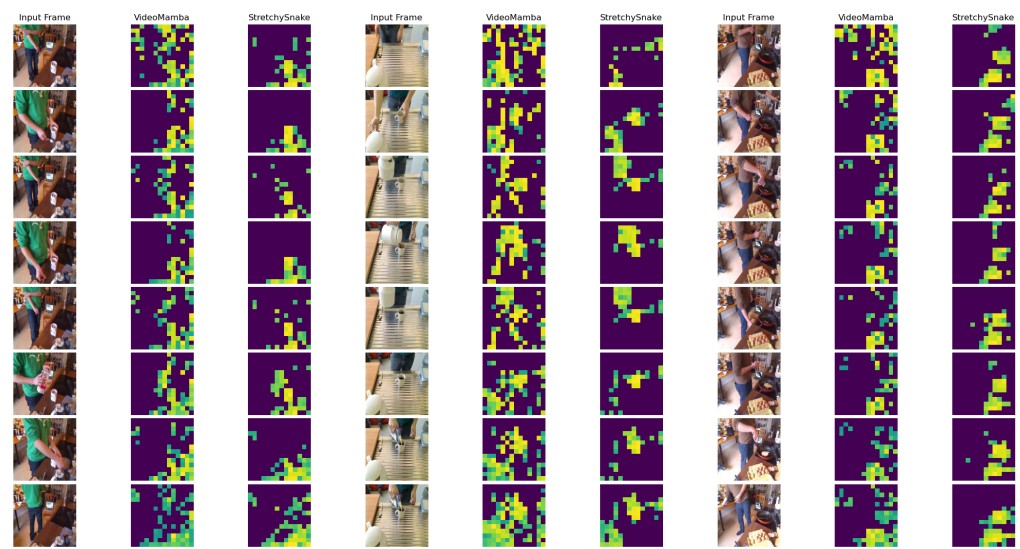

Figure A32: Patch activation map on the Breakfast dataset at $H = W = 224$ pixels.

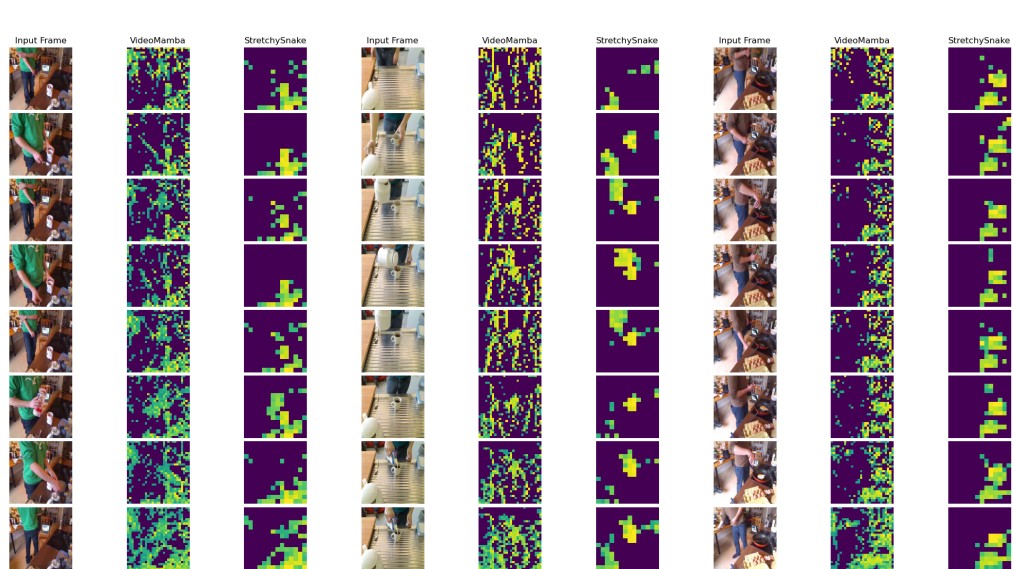

Figure A33: Patch activation map on the Breakfast dataset at $H = W = 448$ pixels.

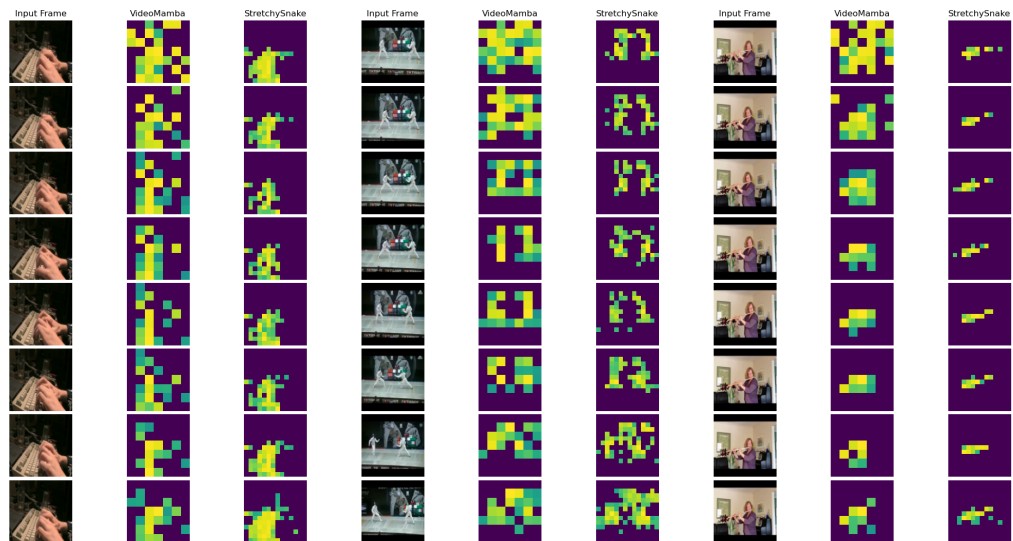

Figure A34: Patch activation map on the UCF-101 dataset at $H = W = 112$ pixels.

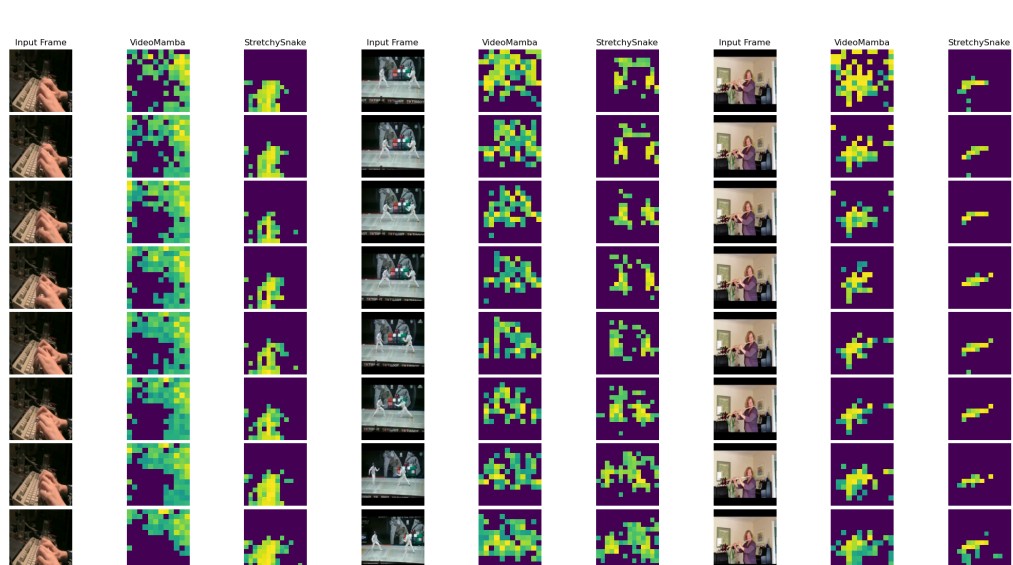

Figure A35: Patch activation map on the UCF-101 dataset at $H = W = 192$ pixels.

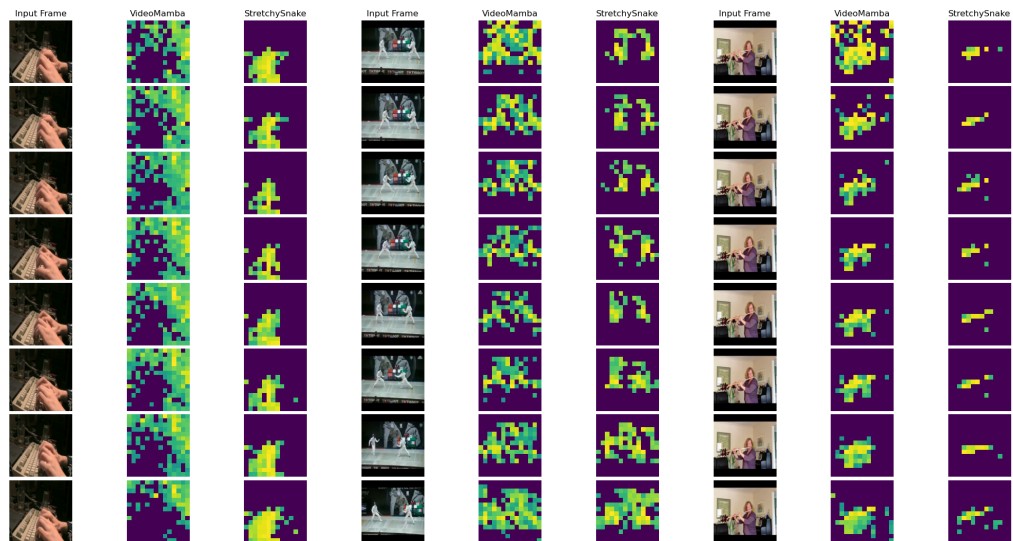

Figure A36: Patch activation map on the UCF-101 dataset at $H = W = 224$ pixels.

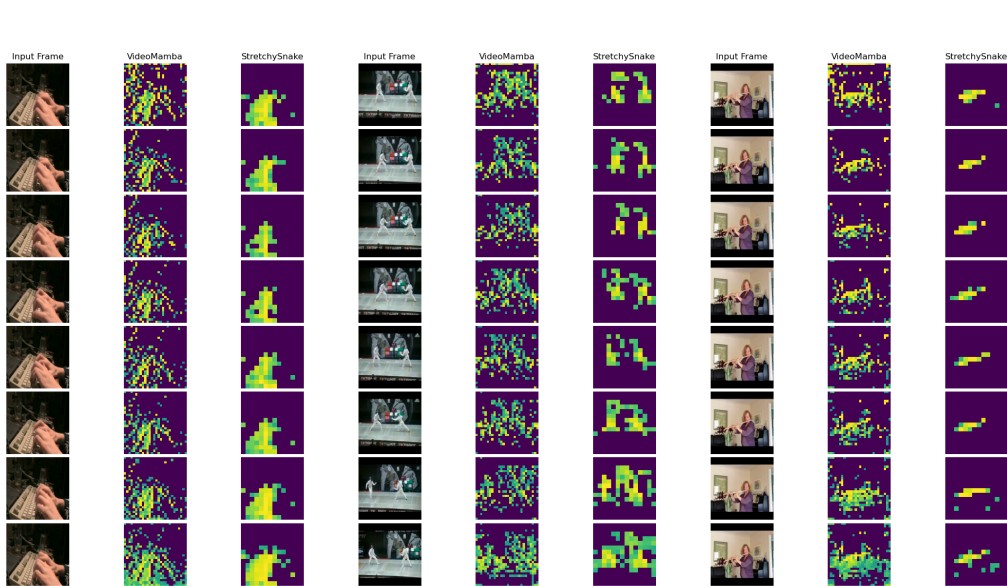

Figure A37: Patch activation map on the UCF-101 dataset at $H = W = 448$ pixels.

