# OpenReview forum: "StretchySnake: Flexible SSM Training Unlocks Action Recognition Across Spatio-Temporal Scales"
_ICLR.cc/2026/Conference — Submitted to ICLR 2026_

### Official Review · Reviewer_P5R8 · 2025-10-26

**Soundness:** 3
**Presentation:** 3
**Contribution:** 3
**Rating:** 6
**Confidence:** 4

**Summary:**

This work proposes StretchySnake for solving the spatio-temporal inflexibility, enabling seamlessly handle videos ranging from short, fine-grained clips to long, complex activities. On 6 action video benchmarks, StretchySnake outperforms vanilla VideoMamba by up to 28%, while simultaneously delivering 3$\times$ speedups and a 90% reduction in GFLOPs.

**Strengths:**

1. The motivation of this work is clear and easy to understand, especially spatio-temporal inflexibility.
2. This work has a detailed theoretical analysis.
3. The workload is full and the experiments are comprehensive.

**Weaknesses:**

1. From most of works for Mamba improvement, they always focus on global modeling of long sequence and computational complexity. This work seems also focus on this. Therefore, I recommend authors highlight their own design and note the corresponding spatio-temporal inflexibility, enhancing their novelty.
2. The writing of the formula is not standardized. For example, conv, concat, and temp in equation 4,5, and 6 are non-variables. It should be written with normal fonts instead of Italic.
3. I hope to see more implementation details, such as how the dataset is processed. Because the SSv2 dataset is direction-sensitive, using a horizontal flip is not appropriate.
4. Given the different capacities of these datasets, are the averages in Table 2 rigorous? Of course, this is just a friendly discussion and does not affect my judgment of this work.
5. The size of Table A3 is too large and looks a little strange.
6. From several figures of t-SNE, the advantages of the proposed method are not well highlighted.

**Questions:**

Please refer to the weaknesses. Overall a good paper.

---

> ### Author Response · Authors · 2025-11-20
>
> >"The writing of the formula is not standardized. For example, conv, concat, and temp in equation 4,5, and 6 are non-variables. It should be written with normal fonts instead of Italic."
>
> Thank you for the suggestion, we have incorporated your changes into the updated pdf with all changes highlighted in blue for transparency and visibility.
>
> >"I hope to see more implementation details, such as how the dataset is processed. Because the SSv2 dataset is direction-sensitive, using a horizontal flip is not appropriate."
>
> Thank you for the insightful comment. For all training and evaluation experiments, we only use two transforms: normalization and resize (to resize every frame to the correct spatial resolution during flexible training/inference) - no other augmentations are used. This is reflected in our code submission, where we provide our dataloaders.
>
> >"Given the different capacities of these datasets, are the averages in Table 2 rigorous?"
>
> Our goal with including the average was to highlight a singular model's ability to perform short-length (UCF-101, HMDB-51), long-length (COIN, Breakfast) and fine-grained (SSv2, Diving-48) action recognition. At a glance, a single model with the highest average performs the best across these three diverse action recognition scenarios, as discussed in Lines 56-80.
>
> >"The size of Table A3 is too large and looks a little strange. From several figures of t-SNE, the advantages of the proposed method are not well highlighted."
>
> Thank you for pointing this out. We have fixed the size of Table A3, and added some additional information to the captions of the t-SNE plots to better highlight the conclusion. Additional discussion can be found in Section A.7.2 as well.

---

### Official Review · Reviewer_mAqy · 2025-10-27

**Soundness:** 2
**Presentation:** 3
**Contribution:** 1
**Rating:** 4
**Confidence:** 4

**Summary:**

This paper proposes STRETCHYSNAKE, a video SSM model with spatio-temporal flexibility (aka st-flexibility) that can be generalized to various spatial and temporal resolutions as well as different patch sizes with improved video understanding performance.
The core contribution is the realization of st-flexibility, which is achieved by interpolating the weights of spatial/temporal positional embeddings and convolution weights during training and inference.
The authors conduct many experiments across different inference configurations and show that st-flexibility is possible and effective in video SSMs.

**Strengths:**

1. This paper is well written and easy to follow.
2. The figures are clear.
3. The authors conduct sufficient experiments to validate the effectiveness of the proposed st-flexibility in video SSMs.
4. Improved performance compared to its counterparts, i.e., the VideoMamba.

**Weaknesses:**

My major concern is the **novelty issue**, because there is a problematic statements regarding existing video models in the core motivation:

L86-L88: *(transformers) reliance on learning explicit token-to-token relationships usually constrains them to fixed input sizes, preventing generalization across diverse spatio-temporal scales.*

**Using 2-D bi-cubic interpolation or 1-D linear interpolation on spatial/temporal positional embeddings and convolution weights (L319-L322) is not first proposed by the authors but a widely adopted technique in video transformers.** Actually, interpolating spatial/temporal and convolutional weights at finetuning or inference stages for various spatio-temporal scales has been a standard practice in the past few years. There are several well-known implementations:

1. TimeSformer: see `forward_features` (L249) at https://github.com/facebookresearch/TimeSformer/blob/main/timesformer/models/vit.py. It exactly uses spatial/temporal interpolation (same as the paper proposed) during inference to adapt for various scales.

2. The timm repo: see `_load_weights` (L1091) at https://github.com/huggingface/pytorch-image-models/blob/main/timm/models/vision_transformer.py. It not only interpolates the weights of the positional embeddings but also the convolution weights, i.e., the `resample_patch_embed` and `resample_abs_pos_embed` functions.

Besides, **the proposed st-flexibility does not really resolve the generability issue because the model is still trapped in the pre-set training scales**, i.e., $R^t=[8,16,32,64]$ and $R^s=[96,128,224,384]$. No experiments are provided to show the performance w/ scales beyond training. For instance, the performance if $R^t=110$ and $R^s=768$.

I believe the above issue is not mean because **frontier video models already reach real generability through Any Resolution techniques, which is totally missing in this paper.** Please see the below papers:

1. Patch n' Pack: NaViT, a Vision Transformer for any Aspect Ratio and Resolution: 1D-Rope for vision understanding.

2. LLaVA-Video: Video Instruction Tuning With Synthetic Data: 1D-Rope for any-scale video understanding.

3. Qwen2-VL: Enhancing Vision-Language Model's Perception of the World at Any Resolution: 2D-Rope for any-scale video understanding.

4. Qwen2.5-VL Technical Report: 3D-Rope for any-scale video understanding.

Besides, **some important baselines that also pre-trained on Kinetics-400 are missing.** For instance, VideoMAE and VideoMAE-V2. Please conduct a rigorous literature review and include them in Table2.

**Questions:**

1. Can you show some results when testing at an unseen scale beyond the pre-set ones, e.g., $R^t=110$ and $R^s=768$?
2. Can you discuss the differences between your method and the pre-developed interpolation ones, e.g., in TimeSformer and timm repos?
3. Please include the latest any-resolution works, e.g., NaViT, LLaVA-Video, Qwen2-VL and Qwen2.5-VL and discuss the merits of your approach over theirs.

---

> ### Author Response · Authors · 2025-11-20
> **Response to Reviewer mAqy (1/4)**
>
> We thank you for your evidently detailed review - your questions are very insightful. In addition to our clarifications in the general comment block at the top, we directly address each of your concerns with additional detail below:
>
> >"**Q1: Can you show some results when testing at an unseen scale beyond the pre-set ones?** The model is still trapped in pre-set training scales. Evaluations beyond training spatio-temporal scales are needed."
>
> * We already show video retrieval results on unseen spatial resolutions (112, 192, 288, 448) across four datasets in Table 1 of the paper. Furthermore, we provide qualitative visualizations of StretchySnake's features and CLS token at unseen resolutions in Sections A.7.2 and A.7.3 in the appendix.
>
> * However, your suggestion to perform evaluation on unseen temporal resolutions is also valuable to supporting our method.  In Tables R2-R5 below, we provide additional video retrieval results across all four coarse-action recognition datasets at the suggested scale of T=110 (number of frames) and H=W=768 (resolution of each frame), along with additional unseen temporal resolutions T = \{4, 24, 128\}. The results follow the same trend in the paper - **StretchySnake retains performance across any spatio-temporal scale, even those unseen during training**, while vanilla VideoMamba degrades in performance.
>
>
> **Table R2.** Comparing StretchySnake and vanilla VideoMamba video retrieval performance on the Breakfast dataset solely on **unseen** spatial and temporal resolutions.
>
> | Model (H=W=768) | T=4  | T=24 | T=110 | T=128 |
> |:----------------:|:----:|:----:|:-----:|:-----:|
> | Vanilla VideoMamba            | 19.4 | 17.5 | 12.7  | 12.4  |
> | StretchySnake             | 45.5 | 46.7 | 45.8  | 48.7  |
>
> **Table R3.** Comparing StretchySnake and vanilla VideoMamba video retrieval performance on the COIN dataset solely on **unseen** spatial and temporal resolutions.
>
> | Model (H=W=768) | T=4  | T=24 | T=110 | T=128 |
> |:----------------:|:----:|:----:|:-----:|:-----:|
> | Vanilla VideoMamba             | 42.9 | 35.9 | 30.7  | 29.3 |
> | StretchySnake              | 50.2 | 52.8 | 50.7  | 55.3  |
>
> **Table R4.** Comparing StretchySnake and vanilla VideoMamba video retrieval performance on the UCF-101 dataset solely on **unseen** spatial and temporal resolutions.
>
> | Model (H=W=768) | T=4  | T=24 | T=110 | T=128 |
> |:----------------:|:----:|:----:|:-----:|:-----:|
> | Vanilla VideoMamba            | 70.6 | 65.4 | 27.4  | 26.7  |
> | StretchySnake              | 90.2 | 92.3 | 90.9  | 90.7  |
>
> **Table R5.** Comparing StretchySnake and vanilla VideoMamba video retrieval performance on the HMDB-51 dataset solely on **unseen** spatial and temporal resolutions.
>
> | Model (H=W=768) | T=4  | T=24 | T=110 | T=128 |
> |:----------------:|:----:|:----:|:-----:|:-----:|
> | Vanilla VideoMamba              | 34.8 | 30.9 | 12.3  | 12.2  |
> | StretchySnake             | 45.3 | 44.0 | 44.0  | 43.7  |
>
>
> >"**Q2: Can you discuss the differences between your method and the pre-developed interpolation ones?** Using 1-D and 2-D interpolations on embeddings is not first proposed by the authors."
>
> * We acknowledge that embedding interpolation has already existed for some models as the reviewer pointed out in certain codebases. However, it is important to note that almost every model uses embedding interpolation as a simple engineering trick during inference to enable the model to ingest any spatial resolution - **this does not instill the model with st-flexibility**.
>
> * The reviewer correctly notes that TimeSFormer has an embedding interpolation function that is only used at inference time. However, **we already show in Section A.2 of the appendix that training TimeSFormer with st-flexibility still yields degraded test-time performance across spatio-temporal resolutions** (with our code provided in the supplementary material). We also trained UniFormer and MViT with st-flexibility in Section A.2 and exhibit the instability and impact of model architecture on the results. See the general comment block and results in Tables R6-R7 below for additional discussion.

---

> ### Author Response · Authors · 2025-11-20
> **Response to Reviewer mAqy (2/4)**
>
> >"**Q3: Please include the latest any-resolution works, e.g., NaViT, LLaVA-Video, Qwen2-VL and Qwen2.5-VL.** Frontier models already leverage multi-resolution techniques (such as AnyRes)."
>
> There are some important distinctions between our work and the works cited by the reviewer. In Tables R6-R7, we provide video retrieval results with the suggested models on one long-action dataset (Breakfast) and one short-action dataset (HMDB-51), with StretchySnake results from the main paper included to highlight the benefit of our method. Detailed discussion for each model is as follows:
>
> * LLaVA-Video uses a rotary embedding (RoPE) in the language model but not for the vision encoder; the vision encoder still uses a fixed, 2-D positional encoding and therefore **cannot perform visual understanding at any scale**. LLaVA-Video encodes videos frame-by-frame using an image encoder (SigLIP [1]) which was statically trained on 384x384 images. To prove its inflexibility in video applications, we perform video retrieval at various spatio-temporal scales with SigLIP in Tables R6-R7. While we understand SigLIP is an image encoder, we want to highlight its drop in performance on video tasks across different spatio-temporal resolutions.
>    * We also want to clarify the difference between our method and the AnyRes technique; AnyRes is only used at inference time, only applied in the spatial dimension by simply taking multiple crops of a high-res image, and is specifically tailored to optimize the number of tokens passed to an LLM. Thus, **AnyRes does not explictly prevent st-inflexbility, but simply reduces token lengths to optimize LLM compute cost during inference.**
>
> * NaViT does apply flexible training to image models (similar to FlexiViT, which we already included in Section 3.3 of the paper), but is limited to image models and tasks. To further exhibit this point, we load a SigLIP model that was trained with the NaViT technique from HuggingFace and perform video retrieval across different spatio-temporal resolutions in Tables R6-R7. We see that **NaViT still does not fully mitigate model inflexibility**, especially across different temporal resolutions in video understanding tasks. We also kindly refer the reviewer to Figures 3 and A13 in the paper, where our st-flexible method significantly outperforms FlexiViT, which is similar to NaViT.
>
> * Qwen2-VL and Qwen2.5-VL are not exactly aligned with our scope of unimodal video action recognition, since the vision encoders used in Qwen-VL family are trained for text-alignment and video question answering. However, your point is still relevant since Qwen2-VL's video encoder [2] is trained with the NaViT technique and RoPE. We extract the vision encoder from Qwen2-VL and Qwen2.5-VL trained with NaViT and perform video retrieval across different spatio-temporal resolutions in Tables R6-R7. The Qwen-VL encoders are ViTs trained for vision-question answering, which is not perfectly suitable for unimodal action recognition/retrieval, however the lack of robustness across spatial and temporal resolutions is still apparent. This shows that **(1) existing image flexibility methods for image models do not translate directly to video understanding tasks**, and more importantly **(2) even video transformer models trained with st-flexibility do not benefit nearly as much as video SSMs**. We believe this is an important finding for the research community to start adopting our SSM-tailored training technique with the growing interest in SSMs.

---

> ### Author Response · Authors · 2025-11-20
> **Response to Reviewer mAqy (3/4)**
>
> **Table R6.** Video retrieval performance on Breakfast across various transformer-based vision encoders and across various spatio-temporal test-time scales. Second column is number of frames used during inference, subsequent columns are spatial resolution, and each model's best accuracy is bolded. StretchySnake results from the main paper are included to better highlight the impact of our best st-flexible training method.
>
> | Model (Vision Encoder)                       | #Frames | 96   | 112  | 128  | 192  | 224  | 288  | 384  | 448  |
> |:----------------------------------------------|:-------:|:----:|:----:|:----:|:----:|:----:|:----:|:----:|:----:|
> | LLaVA -Video (SigLIP)                          |   8     | 11.3 | 10.7 | 14.7 | 15.8 | 14.4 | **21.0** | 17.6 | 15.5 |
> | LLaVA -Video (SigLIP)                          |   16    | 12.2 | 10.4 | 14.7 | 14.7 | 16.1 | 16.4 | **22.8** | 16.3 |
> | NaViT (SigLIP + NaViT)                        |   8     | 13.0 | 14.2 | 14.2 | 9.8  | 20.4 | 22.1 | **25.1** | 24.3 |
> | NaViT (SigLIP + NaViT)                        |   16    | 11.9 | 10.2 | 10.2 | 12.5 | 21.5 | 23.2 | **27.3** | 25.7 |
> | Qwen2 (ViT + NaViT)                           |   8     | 8.0 | 8.8 | 9.0 | 8.8 | 9.5 | 10.2 | **10.8** | 9.3 |
> | Qwen2 (ViT + NaViT)                           |   16    | 8.3 | 9.0 | 9.3 | 9.0 | 9.8 | 10.7 | **10.8** | 9.9 |
> | Qwen2.5 (ViT + NaViT)                         |   8     | 9.2 | 9.9 | 10.5 | 10.4 | 10.2 | 12.5 | **12.8** | 11.9 |
> | Qwen2.5 (ViT + NaViT)                         |   16    | 9.5 | 10.0 | 10.2 | 10.6 | 10.0 | 13.0 | **13.1** | 12.7 |
> | VideoMamba (VideoMamba)                                |   8     | 22.0 | 23.1 | 24.9 | 31.9 | **43.2** | 40.7 | 34.5 | 30.5 |
> | StretchySnake (VideoMamba + Static-Tokens)    |   8     | 49.4 | 50.0 | 49.7 | 49.1 | **53.7** | 52.8 | 51.4 | 47.7 |
> | VideoMamba (VideoMamba)                                |   16    | 22.0 | 22.9 | 22.0 | 37.5 | 41.8 | **42.1** | 33.3 | 33.1 |
> | StretchySnake (VideoMamba + Static-Tokens)    |   16    | 49.4 | 49.2 | 48.3 | 50.3 | **53.4** | 52.5 | 48.6 | 50.3 |
>
>
>
> **Table R7.** Video retrieval performance on HMDB-51 across various transformer-based vision encoders and across various spatio-temporal test-time scales. Second column is number of frames used during inference, subsequent columns are spatial resolution, and each model's best accuracy is bolded. StretchySnake results from the main paper are included to better highlight the impact of our best st-flexible training method.
>
> | Model (Vision Encoder)                       | #Frames | 96  | 112 | 128 | 192 | 224 | 288 | 384 | 448 |
> |:----------------------------------------------|:-------:|:---:|:---:|:---:|:---:|:---:|:---:|:---:|:---:|
> | LLaVA -Video (SigLIP)                          |   8     | 7.7 | 10.6 | 8.6 | 12.8 | 15.7 | 18.5 | **40.2** | 18.1 |
> | LLaVA -Video (SigLIP)                          |   16    | 8.6 | 9.8 | 9.9 | 11.9 | 16.3 | 20.5 | **40.5** | 19.1 |
> | NaViT (SigLIP + NaViT)                        |   8     | 8.7 | 11.2 | 10.8 | 13.8 | 18.9 | 22.7 | **47.5** | 23.3 |
> | NaViT (SigLIP + NaViT)                        |   16    | 9.3 | 11.2 | 12.2 | 12.5 | 21.5 | 23.2 | **27.3** | 25.7 |
> | Qwen2 (ViT + NaViT)                           |   8     | 7.0 | 7.6 | 7.8 | 7.4 | 9.5 | 10.2 | **10.8** | 9.9 |
> | Qwen2 (ViT + NaViT)                           |   16    | 7.7 | 8.3 | 8.8 | 8.9 | 9.2 | 10.8 | **11.6** | 10.8 |
> | Qwen2.5 (ViT + NaViT)                         |   8     | 8.0 | 8.8 | 9.6 | 9.8 | 9.8 | 12.0 | **12.1** | 11.1 |
> | Qwen2.5 (ViT + NaViT)                         |   16    | 8.2 | 8.8 | 9.4 | 10.0 | 10.1 | 12.2 | **12.3** | 11.3 |
> | VideoMamba (VideoMamba)                                |   8     | 36.5 | 44.4 | 49.1 | 57.8 | **58.7** | **58.7** | 55.3 | 52.0 |
> | StretchySnake (VideoMamba + Static-Tokens)    |   8     | 61.6 | 62.7 | 63.2 | **64.2** | 63.2 | 62.9 | 62.1 | 62.2 |
> | VideoMamba (VideoMamba)                                |   16    | 35.0 | 42.8 | 49.8 | 56.5 | **58.2** | 57.8 | 53.7 | 51.6 |
> | StretchySnake (VideoMamba + Static-Tokens)    |   16    | 60.6 | 63.3 | 63.6 | **64.4** | 63.0 | **64.4** | 64.0 | 62.1 |

---

> > ### Author Response · Authors · 2025-11-20
> > **Response to Reviewer mAqy (4/4)**
> >
> > >"Other baseline models trained on K400 are missing."
> >
> > Thank you for the suggestion! We have included video retrieval results in Table R8 below on VideoMAE and VideoMAEv2. Note that VideoMAEv2 is trained with a mixture of K400, K600, and K700 data, and that both models are older than some models already included in Table 2 of the main paper. Regardless, the results in Table R8 exhibits that StretchySnake performs the best on average across all 6 datasets, meaning it has the best generalization capabilities across short, long, and fine-grained action recognition alike due to our st-flexible training method.
> >
> >
> > **Table R8.** Video retrieval performance of VideoMAE and VideoMAEv2 across the 6 datasets used in the main paper. StretchySnake results from the main paper are included for  comparison.
> >
> > | Model | UCF101 | HMDB51 | COIN | Breakfast | SSV2 | Diving48 | Average |
> > |:----------------:|:------:|:------:|:----:|:----------:|:----:|:--------:|:-------:|
> > | **VideoMAE** [3]  | 93.1 | 60.6 | 66.0 | 43.9 | 9.8 | 8.3 | 47.0 |
> > | **VideoMAE-V2** [4]  | 95.2 | 64.0 | 52.0 | 50.0 | 12.1 | 14.9 | 48.0 |
> > | **StretchySnake** | **94.5** | **66.1** | **80.0** | **60.2** | **12.4** | **15.1** | **54.7** |
> >
> > We thank the reviewer again for their detailed review which we believe adds additional rigor to our reported results. We hope that we have addressed all concerns, and are happy to add provide additional clarification if needed!
> >
> > ---
> >
> > [1] Zhai, X., Mustafa, B., Kolesnikov, A., \& Beyer, L. (2023). Sigmoid loss for language image pre-training. In Proceedings of the IEEE/CVF international conference on computer vision (pp. 11975-11986).
> >
> > [2] Wang, P., Bai, S., Tan, S., Wang, S., Fan, Z., Bai, J., ... & Lin, J. (2024). Qwen2-vl: Enhancing vision-language model's perception of the world at any resolution. arXiv preprint arXiv:2409.12191.
> >
> > [3] Tong, Z., Song, Y., Wang, J., & Wang, L. (2022). Videomae: Masked autoencoders are data-efficient learners for self-supervised video pre-training. Advances in neural information processing systems, 35, 10078-10093.
> >
> > [4] Wang, L., Huang, B., Zhao, Z., Tong, Z., He, Y., Wang, Y., ... & Qiao, Y. (2023). Videomae v2: Scaling video masked autoencoders with dual masking. In Proceedings of the IEEE/CVF conference on computer vision and pattern recognition (pp. 14549-14560).

---

### Official Review · Reviewer_35wV · 2025-10-30

**Soundness:** 2
**Presentation:** 3
**Contribution:** 2
**Rating:** 4
**Confidence:** 4

**Summary:**

On the recent progresses of SSMs image classification, video understanding and 3D vision, this paper aims to adapt the recent SSM and training to fine-grained action recognition. It implements and incrementally improves the training on video learning and representation on fine-grained action recognition benchmarks. The purposes of the experiments are to identify good type of st-flexibility of SSM for video representation on fine-grained action videos, demonstrate the improvement over vanilla baseline, and compare to SOTA on limited training dataset. The reported progresses have shown a certain level of extension of existing SSMs on additional tasks.

**Strengths:**

Investigating to apply recent progresses on SSM to video representation and fine-grained action recognition task.

**Weaknesses:**

The novelty of the paper is still weak, and the evaluations are not completed and concrete enough to show the significance of the progresses. First, the technical descriptions present the existing approaches and implementation details. Not clear what are novel method, model architecture, or learning function and algorithms beyond existing approaches. This paper presents a few incremental progresses and extension of existing SSMs. May need to focus on deeper study and big jump to show significant progresses. Second, the purposes of the experiments are not clear and strong. To show the optimal type of st-flexibility for SSMs in a concrete background, experiments on a wide range of vision tasks would be convincing. In the paper, only experiment for video retrieval on 4 action recognition datasets, lack formal benchmarking on related research topics. On second experiment, only compared with one baseline model. On the third part, the protocol of benchmarking is unclear. If it wants to focus on performance on OOD datasets of fine-grained action recognition with limited training set, it may need to follow a formal protocol and comparison to SOTA on leaderboard benchmarks.

**Questions:**

What are the novel model modules or learning approaches beyond existing SSM technology proposed in this paper? What are the formal protocol, training set and evaluation sets used in the previous benchmarking in Table 2?

---

> ### Author Response · Authors · 2025-11-20
>
> Thank you for your review! Our general comment block at the top provides more clarity on the novelty and contribution of our work. However, we directly answer your questions below as well.
>
>
> >"Not clear what are novel method, model architecture, or learning function and algorithms beyond existing approaches, This paper presents a few incremental progresses and extension of existing SSMs."
>
> We kindly refer to the general comment block, where we discuss in detail how our work separates itself from existing approaches. We also discuss preliminary information, motivation for our work, and detail our full methodology in Section 3.1, Section 3.2, and Section 3.3, respectively. Figure 2 also provides a high-level overview of our novel training method.
>
> >"The purposes of the experiments are not clear and strong. To show the optimal type of st-flexibility for SSMs in a concrete background, experiments on a wide range of vision tasks would be convincing. In the paper, only experiment for video retrieval on 4 action recognition datasets, lack formal benchmarking on related research topics"
>
> We apologize for the confusion, we should note that we aim to identify the best st-flexible method for video SSMs **for action recognition**. We have added that clarification with blue text in the updated draft. We also refer to Section A.4 and Table A1 in our appendix, where we show that StretchySnake outperforms vanilla VideoMamba on the Long Video Understanding dataset, which is not related to action recognition. We included this experiment as a proof of concept that our method shows potential in other video understanding tasks besides action recognition.
>
>
> >"On second experiment, only compared with one baseline model"
>
> As we mentioned in Lines 252-253, VideoMamba is the only published SSM-based video model available at the time of writing. This is simply a result of how new video SSMs are in the field, yet we further discuss in Section 3.2 why our method is compatible with any future video SSM. We also see the relative infancy of SSM's as a supporting factor to the novelty of our work, as future SSM works can leverage our unique training method if published.
>
> >"[In Table 2], the protocol of benchmarking is unclear. If it wants to focus on performance on OOD datasets of fine-grained action recognition with limited training set, it may need to follow a formal protocol and comparison to SOTA on leaderboard benchmarks."
>
> We discuss in Section A.3 why we use the evaluation protocols present in the paper. We follow previous works cited in Section A.3 that use video retrieval, linear probing, and full-finetuning to investigate the quality of learned representations in trained models.
>
> We look forward to hearing from you and if we can offer any additional clarifications, thank you.

---

### Author Response · Authors · 2025-11-20
**General Rebuttal Comments**

To address some common concerns across reviewers, we provide clarification on 2 key main contributions of our work. We address each reviewer's specific questions with additional details in our separate responses below.


1. Novelty
    * Our first point of novelty is instilling st-flexibility **during training**, specifically for video SSMs. We do not claim embedding interpolation itself as our own work, but rather provide extensive experiments and analyses to prove that these differentiable interpolations can be used during training to improve action recognition in video SSMs. We believe the breadth of our experiments is a contribution to the research community - our evaluations across 6 different action recognition datasets (Section 4), 3 evaluation protocols (video retrieval, linear probing, full-finetuning, Section 4.3), types of transformer models (Section A.2), additional video understanding tasks (Section A.4), and ablations (Section A.6) demonstrates that st-flexible training works for SSMs, and more importantly its significant benefits are overlooked in current video SSM practices. We provide dataset details in Table R1 to highlight StretchySnake's ability to adapt to various action recognition settings.

    * We are the first to show that st-flexible training provides substantially larger gains for video SSMs than for video transformers, a distinction not yet examined in prior work. In fact, this point may have been previously obfuscated by the fact that **naively applying st-flexible methods to video transformer models does not yield large performance gains** (see Section A.2 and further discussion in our rebuttal below). The glaring benefit for video SSMs is made especially clear in Table 1 of the paper, where StretchySnake outperforms vanilla VideoMamba on all spatio-temporal scales (by up to 28%), including VideoMamba's own default configuration (Section 4.2). We include further evidence of this effect in Tables R2-R5 in our rebuttal.

2. Additional Comparisons to Image Methods
     * Reviewer mAqy pointed out that there are some existing works that either (a) apply embedding interpolations during inference, or (b) perform image flexible training for image models (see Section 2.3, main paper). We highlight the contribution of our work by **exhibiting the failures of applying these flexible image training methods to video understanding tasks in Tables R6-R7 of our rebuttal below**.




**Table R1.** Dataset details for each dataset used in the paper. StretchySnake can adapt to short-form, long-form, and fine-grained action recognition scenarios as a singular model.

| **Dataset** | **Video Lengths** | **Label / Action Type** | **Dataset Type** | **Fine-grained?** |
|-------------|-------------------|-------------------------|------------------|-------------------|
| **UCF-101** | ~7 seconds | 101 sports / daily-life actions | Short-action, coarse-grained | No |
| **HMDB-51** | ~3-6 seconds | 51 human actions from movies / YouTube | Short-action, coarse-grained | No |
| **Breakfast** | ~120–180 seconds | 48 cooking actions in long activity sequences | Long-action, procedural activities | No |
| **COIN** | ~150–300 seconds | 180 instructional task steps | Long-action, instructional activities | No |
| **SSv2 (Something-Something V2)** | ~4 seconds | 174 fine-grained object interactions | Fine-grained, temporal reasoning | Yes |
| **Diving-48** | ~3–6 seconds | 48 subtle dive types requiring fine motion cues | Fine-grained, high-speed sports | Yes |

---

### Author Response · Authors · 2025-12-03
**Summary**

We would like to thank all reviewers, area chairs, and program chairs for their time and effort throughout this unusual review cycle. Below is a concise summary of our contributions and the main points addressed in our rebuttal.

### **Contribution**
We are the first to propose **spatio-temporal flexible video training for state space models (SSMs)**. By dynamically interpolating input video resolution, patch size, and positional embeddings during training, we show that SSMs learn substantially more generalized representations - yielding large gains on unseen data (up to **28%**) and consistent improvements when performing linear probing/full-finetuning.

Because flexibility can be implemented in multiple ways, we introduce and evaluate **five different variants** to identify the most effective formulation. Our resulting flexible SSM, **StretchySnake**, is validated across **six action-recognition datasets**, spanning 3 types of action recognition (short-action, long-action, fine-grained), and **three evaluation protocols** (video retrieval, linear probing, full fine-tuning). We beat VideoMamba across every dataset, evaluation protocol, and spatio-temporal scale, including many existing video transformer models.

Importantly, with how recent video SSMs are, they are currently trained the same as transformers with no prior works leveraging the unique characteristics of SSMs. We reveal a new and valuable insight: **SSMs benefit dramatically more from spatio-temporal flexible training than transformer-based models**. We support this claim with theoretical motivation and extensive empirical evidence throughout the paper (Section 3.2, Section A.2) and hope that our method will foster further adoption and research efforts towards video SSMs.

### **Rebuttal**

**Reviewer P5R8** gave a positive score and highlighted the clarity, theoretical motivation, and comprehensive experiments of our work. We addressed some of their minor formatting concerns in the updated draft.

**Reviewer 35wV** may have misunderstood several aspects of our method. Their summary included inaccuracies, and several of their listed weaknesses are already addressed in the paper. Regardless, we provided clarification on these points in our rebuttal, but no further discussion occurred before the response window closed.

**Reviewer mAqy** raised some insightful questions. Their key concerns and our responses are summarized below:

- **Concern: Novelty may be limited because prior work uses embedding interpolation.**
  - Prior works apply embedding interpolation only *at test time*. We perform it *during training* for video action recognition and specifically for **SSMs**. Although the reviewer mentions TimeSFormer, it relies on interpolation only at inference, and - as shown in Section A.2 - cannot learn spatio-temporal flexibility during training. We view the intuitiveness of our method as a strength. Combined with the large performance gains, extensive experiments and ablations, and our novel analysis comparing flexibility in transformers versus SSMs, we believe our work offers a meaningful contribution to the research community.

- **Concern: Results on unseen spatio-temporal scales are not provided.**
  - Our main paper includes both quantitative and qualitative results on unseen resolutions (see Table 1, also Section A.7). We added more quantitative results during the rebuttal for completeness.

- **Concern: Other flexible image-training methods exist.**
  - We acknowledged these references in our rebuttal and demonstrated that such methods yield limited gains for transformers (as we conclude in Section A.2), while SSMs benefit strongly. We provided results for all models requested by the reviewer in Tables R6 and R7 of our response. We also note that we already compare against FlexiViT in the main paper, which is similar to the references brought up by the reviewer.

- **Concern: Some smaller baselines were missing.**
  - We added these baselines during the rebuttal, however they underperform the baselines already provided in the paper.


We would also like to note that our entire codebase is provided in the supplementary for full reproducibility.

---

### Meta-Review · Area_Chair_d5fr · 2025-12-29

**Summary:**

The strengths of the paper are highlighted consistently by the reviewers. Reviewer 35wV writes that the paper is “investigating to apply recent progresses on SSM to video representation and fine-grained action recognition task,” and notes that the work “aims to adapt the recent SSM and training to fine-grained action recognition” with “a certain level of extension of existing SSMs on additional tasks.” Reviewer mAqy states that “this paper is well written and easy to follow,” that “the figures are clear,” that “the authors conduct sufficient experiments to validate the effectiveness of the proposed st-flexibility in video SSMs,” and that there is “improved performance compared to its counterparts, i.e., the VideoMamba.” Reviewer P5R8 summarizes that StretchySnake “solves the spatio-temporal inflexibility, enabling seamlessly handle videos ranging from short, fine-grained clips to long, complex activities,” and lists as strengths that “the motivation of this work is clear and easy to understand, especially spatio-temporal inflexibility,” that “this work has a detailed theoretical analysis,” and that “the workload is full and the experiments are comprehensive.”

The main weaknesses concern novelty, clarity of the unique contribution beyond existing practice, and the completeness and positioning of the evaluation. Reviewer 35wV states that “the novelty of the paper is still weak, and the evaluations are not completed and concrete enough to show the significance of the progresses,” noting that “this paper presents a few incremental progresses and extension of existing SSMs” and that it is “not clear what are novel method, model architecture, or learning function and algorithms beyond existing approaches.” Reviewer 35wV also finds that “the purposes of the experiments are not clear and strong” and argues that to “show the optimal type of st-flexibility for SSMs in a concrete background, experiments on a wide range of vision tasks would be convincing,” while criticizing that “only experiment for video retrieval on 4 action recognition datasets” and that “on the third part, the protocol of benchmarking is unclear.” Reviewer mAqy’s major concern is “the novelty issue,” pointing out that “using 2-D bi-cubic interpolation or 1-D linear interpolation on spatial/temporal positional embeddings and convolution weights is not first proposed by the authors but a widely adopted technique in video transformers,”  and arguing that “the proposed st-flexibility does not really resolve the generability issue because the model is still trapped in the pre-set training scales.” Reviewer mAqy also notes that “frontier video models already reach real generability through Any Resolution techniques” such as NaViT, LLaVA-Video, Qwen2-VL, Qwen2.5-VL and points to “some important baselines that also pre-trained on Kinetics-400” that are missing. Reviewer P5R8, while positive overall, suggests that “this work seems also focus on” global modeling and complexity “from most of works for Mamba improvement” and recommends that the authors “highlight their own design and note the corresponding spatio-temporal inflexibility, enhancing their novelty,” and raises several minor issues about formula notation, implementation details (e.g., SSv2 flips), averaging across datasets, and t-SNE figures.

In terms of ratings, the reviewers recommend 4, 4, 6 scores. the AC recommends rejection because the overall balance of reviews, and the remaining concerns after rebuttal, suggest that the work falls short of the bar for ICLR in this current form. Reviewer 35wV characterizes the paper with unclear novelty and experimental purposes, and emphasizes the lack of broader formal benchmarking beyond video retrieval on a limited set of action recognition tasks. Reviewer mAqy’s central point is that the core techniques(spatial/temporal embedding interpolation and convolution weight resampling) are “a widely adopted technique in video transformers,” and that any‑resolution vision encoders already exist; even after the authors’ detailed comparisons, the distinction between a new training recipe for SSMs and prior flexible training/inference practice remains somewhat narrow. Reviewer P5R8 is positive but not strongly so (would not mind if paper is rejected) and encourages the authors to “highlight their own design and note the corresponding spatio-temporal inflexibility, enhancing their novelty,” underscoring that _even the supportive reviewer_ feels that the novelty needs to be better asserted. The rebuttal substantially strengthens the empirical evidence,adding results at unseen scales, comparisons to TimeSFormer/FlexiViT/NaViT/LLaVA‑Video/Qwen2‑VL, and VideoMAE baselines, and clarifies the scope and implementation, but in the AC’s view it does not entirely resolve the perception that the main idea is an intuitive, incremental training strategy built from known ingredients, albeit applied very thoroughly and effectively in the SSM video setting. With two reviewers remaining marginally below the acceptance threshold and only one marginally above, and with the key novelty concerns only partially addressed, the AC does not see a strong enough consensus to recommend acceptance.
On balance, the AC sees not basis to overturn the reviewer suggestions and also recommends rejection. The paper presents a clear and well-motivated training recipe that uses spatio-temporal flexible sampling and interpolation to make VideoMamba-style SSMs more robust across resolutions and sequence lengths, and it shows good gains over VideoMamba across six action datasets and multiple protocols; however, several reviewers continue to see the contribution as an incremental application of widely used interpolation techniques, with limited architectural or algorithmic novelty and somewhat narrow formal benchmarking beyond action recognition. The AC highly recommends the authors to address the concerns of the reviewers and take into account their suggestions of improvement when preparing a revised version.

**Reviewer Concerns:**

The rebuttal addresses a number of specific reviewer concerns well, but does not completely resolve the core novelty questions. For Reviewer 35wV, the authors emphasize that their novelty lies in instilling “st-flexibility during training, specifically for video SSMs,” and point to Sections 3.1–3.3 and Figure 2 as detailed descriptions of their training method. They also clarify that their goal is “to identify the best st-flexible method for video SSMs for action recognition,”not all vision tasks, and mention additional evidence on a long video understanding dataset in Section A.4, and explain that VideoMamba is the only published video SSM baseline currently available. These responses clarify intention and scope, but likely do not fully change Reviewer 35wV’s perception that the work is “incremental” extension of existing SSMs with limited methodological novelty. For Reviewer mAqy, the authors provide extensive additional material: they show results at unseen scales (e.g., T=110, H=W=768, and other T values) in Tables R2–R5, where StretchySnake retains performance and VideoMamba degrades; they explicitly discuss differences between their method and pre‑developed interpolation in TimeSFormer and timm, arguing that prior work uses interpolation only at inference and that training transformers with st‑flexibility does not yield similar gains; and they add comparisons against NaViT‑style and AnyRes‑style models (LLaVA‑Video SigLIP, NaViT, Qwen2‑VL encoders) in Tables R6–R7, showing weaker flexibility on video retrieval. They also include new baselines VideoMAE and VideoMAE‑V2 in Table R8, and report that StretchySnake is best on average across six datasets. These additions address many of the concrete requests (unseen scales, missing baselines, discussion of any-resolution models), and could plausibly move Reviewer mAqy towards a slightly higher view of the contribution; however, the basic observation that “using 2-D bi-cubic interpolation or 1-D linear interpolation” is not new remains, and the authors frame their novelty as how these known interpolations are used for SSM training rather than as a fundamentally new mechanism. For Reviewer P5R8, the authors fix notation, explain that they do not use horizontal flips on SSv2 and provide minimal augmentation, justify the use of averages in Table 2, and adjust Table A3 and t‑SNE captions. These concerns are minor should be largely resolved now. Overall, the rebuttal significantly improves clarity and responds thoroughly to specific questions, but does not fully eliminate the reviewers’ doubts about the conceptual leap beyond existing flexible training / interpolation techniques.

**Reviewer Scores:**

Because reviewers did not have the chance to update their ratings after the rebuttal, we can only speculate how they might have reacted. Reviewer 35wV might appreciate the clarifications about scope and the added long‑video experiment, but their main statements about weak novelty and incomplete evaluation are likely only partly addressed; it is plausible their score would remain around 4. Reviewer mAqy(also at 4) explicitly requested unseen-scale results and additional baselines, which the authors provided in detail; they might increase their assessment somewhat, but the key novelty concern around interpolation being standard practice and any‑resolution models existing remains, so a dramatic shift is unlikely. Reviewer P5R8(at rating of 6), saw the paper as a good contribution already and raised only minor issues, which are addressed; this reviewer would probably maintain or slightly strengthen their positive stance. Thus, after rebuttal, the likely configuration would still be one clearly positive but not strongly convincing review (P5R8) and two marginally negative reviews with improved but still somewhat unresolved concerns about novelty and positioning (35wV + mAqy).

---

### Decision · Program_Chairs · 2026-01-26

Reject